



# Airborne SnowSAR data at X- and Ku- bands over boreal forest, alpine and tundra snow cover

Juha Lemmetyinen[1], Juval Cohen[1], Anna Kontu[1], Juho Vehviläinen[1], Henna-Reetta Hannula[1], Ioanna Merkouriadi[1], Stefan Scheiblauer[2], Helmut Rott[2], Thomas Nagler[2], Elisabeth Ripper[2], Kelly Elder[3], Hans-Peter Marshall[4], Reinhard Fromm[5], Marc S. Adams[5], Chris Derksen[6], Joshua King[6], Adriano Meta[7], Alex Coccia[7], Nick Rutter[8], Melody Sandells[8], Giovanni Macelloni[9], Emanuele Santi[9], Marion Leduc-Leballeur[9], Richard Essery[10], Cecile Menard[10] and Michael Kern[11]

[1]Finnish Meteorological Institute, Helsinki, Finland
[2]ENVEO IT GmbH, Innsbruck, Austria
[3]Rocky Mountain Research Station, US Forest Service, Fort Collins, CO, USA
[4]Department of Geosciences, Boise State Univ., Boise, ID, USA
[5]Department of Natural Hazards, Austrian Research Centre for Forests (BFW), Innsbruck, Austria
[6]Environment and Climate Change Canada, Climate Research Division, Toronto, Ontario, M3H5T4, Canada
[7]Metasensing BV, The Netherlands
[8]Northumbria University, Newcastle upon Tyne, UK
[9]Institute of Applied Physics "Nello Carrara", Florence, Italy
[10]School of Geosciences, University of Edinburgh, Edinburgh, UK
[11]European Space Research and Technology Center, European Space Agency

*Correspondence to*: For Sodankylä and Saariselkä data, Juha Lemmetyinen (juha.lemmetyinen@fmi.fi); for AlpSAR data, Helmut Rott (helmut.rott@enveo.at); for TVC data, Joshua King (joshua.king@canada.ca ).

**Abstract.** The European Space Agency SnowSAR instrument is a side looking, dual polarized (VV/VH), X/Ku band synthetic aperture radar (SAR), operable from a small aircraft. Between 2010 and 2013, the instrument was deployed at several sites in Northern Finland, Austrian Alps, and northern Canada. The purpose of the airborne campaigns was to measure the backscattering properties of snow-covered terrain to support the development of snow water equivalent retrieval techniques using SAR. SnowSAR was deployed in Sodankylä, Northern Finland for a single flight mission in March 2011 and twelve missions at two sites (tundra and boreal forest) in the winter of 2011-2012. Over the Austrian Alps, three flight missions were performed between November 2012 and February 2013 over three sites located in different elevation zones, representing a montane valley, Alpine tundra, and a glacier environment. In Canada, a total of two missions were flown in March and April 2013, over sites in the Trail Valley Creek watershed, Northwest Territories, representative of the tundra snow regime. This paper introduces the airborne SAR data, as well as coincident in situ information on land cover, vegetation and snow properties. To facilitate easy access to the data record the datasets described here are deposited in a permanent data repository (https://doi.pangaea.de/10.1594/PANGAEA.933255; Lemmetyinen et al., 2021).

A temporary link to access the data without login information is provided for reviewers of this manuscript:

https://www.pangaea.de/tok/e8c562c3c8a15ac34daa83d00c76fcb347330884



# 1  Introduction

The amount of water stored in seasonal snow is an essential natural resource (Sturm et al., 2017). The duration and extent of snow cover, which can be monitored with relative ease using spaceborne sensors, show notable reductions over the timescale of satellite observations (e.g. Hori et al., 2017). Modeling studies and satellite data records, supported by ground observation networks, have provided indications of decreasing trends in snow water equivalent (SWE) over large extents of the Northern Hemisphere (Liston and Hiemstra 2011; Pulliainen et al., 2020). However, current satellite SWE products still lack sufficient spatial resolution and precision to meet observational requirements of many hydrological applications (Mudryk et al., 2015; Larue et al., 2017). Radar sensors, operating at sufficiently high frequencies, have been proposed to meet these requirements. The proposed Cold Regions Hydrology High-Resolution Observatory (CoReH2O) satellite mission aimed to retrieve SWE over land at a spatial resolution of 200 -500 m, at 3-15 day temporal resolution, with a precision better than 3 cm for SWE $\leq$ 30 cm and better than 10 % for SWE > 30 cm (Rott et al., 2010). The sensor proposed for CoReH2O was a dual polarization Synthetic Aperture Radar (SAR) operating at X- and Ku-bands. While CoReH2O was not selected for mission implementation, Phase A studies motivated the acquisition and analysis of airborne radar data to advance the scientific readiness.

The European Space Agency (ESA) SnowSAR instrument is a dual-polarization (VV, HV), airborne side looking SAR measuring at X- and Ku-bands (Coccia et al., 2011). Between 2010 and 2013, the instrument was operated at several sites in northern Finland, Austrian Alps, and northern Canada. The purpose of the airborne campaigns was to gather information on backscattering properties of snow-covered terrain in support of CoReH2O mission studies, measuring over a range of climatological snow classes and land cover regimes. As a part of the Nordic Snow Radar Experiment (NoSREx, Lemmetyinen et al., 2016), SnowSAR was deployed in Sodankylä, Northern Finland for a single flight in March 2011 and a total of ten acquisition flights in the winter of 2011-2012. Two additional acquisitions took place over a tundra site near the town of Saariselkä, Finland. Three flight missions were performed between November 2012 and February 2013 in the AlpSAR campaign over three sites located in different elevation zones of the Austrian Alps, representing a montane valley (Leutasch), Alpine tundra (Rotmoos), and a glacier environment (Mittelbergferner) (Rott et al., 2013). In Canada, the TVCEx campaign took place in March and April 2013, with two flight campaigns over sites in the Trail Valley Creek (TVC) watershed, Northwest Territories, representative of the tundra snow regime.

Data from SnowSAR are provided as calibrated and geocoded normalized radar cross sections (sigma-nought, $\sigma^0$). At a flight altitude of 1200 meters above ground, the SnowSAR swaths span approximately 400 meters in ground range and up to several kilometres in the flight direction, while the calibrated $\sigma^0$ values are provided at spatial resolutions of 2 m and 10 m. Radiometric calibration was performed by a combination of: 1) internal calibration, 2) corner reflector targets installed along flight paths, and 3) the application of TerraSAR-X imagery and tower-based radar observations (for cross-polarized backscatter; see section 2.3). Based on repeated overpasses and cross-comparison with TerraSAR-X, the absolute accuracy at X-band was determined





to be better than 1 dB and the radiometric stability better than 1 dB at both X and Ku bands for 95 % of the data (Di Leo et al., 2016).

While data from individual sites have already been used in scientific studies (Cohen et al., 2015; Montomoli et al., 2015; King et al., 2018; Zhu et al., 2018; Santi et al., in review), an effort was made here to consolidate analyses by providing both the airborne SAR acquisitions and supporting ancillary data for all sites in a common format. Consistent processing steps were applied to geocode and classify SAR data according to land cover and snow regime. Supporting information such as

topography, vegetation cover and forest properties were added to the database where available. Accordingly, concurrent in situ snow and meteorological data provide means for analysis of SnowSAR backscatter variability in different environmental circumstances. To reduce sensitivity to radar speckle, increase the equivalent number of looks (ENL), and to minimize local-scale sensitivity to vegetation, all gridded data were resampled to 10 m spatial resolution; NetCDF data packages of SnowSAR and ancillary data were produced for each site. The main output is the classified database of observations including:

- SnowSAR backscattering for both frequencies and co- and cross-pol channels: KuVH, KuVV, XVH and XVV
- SnowSAR standard deviation of original observations inside aggregated grid cells for channels: KuVH, KuVV, XVH and XVV
- SnowSAR local incidence angle for channels: KuVH, KuVV, XVH and XVV

- Ancillary data on topography (Digital Elevation Model, DEM), vegetation/forest properties and land cover classification (depending on the availability) geocoded to the same grid as SnowSAR data.
- In situ snow and meteorological properties (manual data collection and automated stations)

Section 2 of this manuscript describes the SnowSAR instrument including estimated calibration accuracy. Sites, including a summary of collected SAR imagery and in situ observations, are described in Section 3. Data processing and the contents of

95 the data repository are described in Section 4. Access to the data repository is described in Section 5; Section 6 discusses practical issues related to the use of these data and Section 7 provides a short summary.

## 2    SnowSAR instrument

### 2.1    System specifications

The SnowSAR instrument was developed in 2010-2011 to support Phase A studies of the CoReH2O satellite mission through simultaneous measurements of X- and Ku-band radar backscatter from an airborne platform. The instrument was developed by MetaSensing B.V., to emulate the anticipated observational capabilities of the CoReH2O sensor (center frequencies, polarization diversity) and to be operable from a small aircraft. The first version of the instrument was flown in Sodankylä,



Finland, in March 2011. After the first flight, changes were made to the instrument configuration including upgraded antennas. The main characteristics of the system are summarized in Table 1.

**Table 1: Main characteristics of SnowSAR system.**

| System parameter | Performance | Comments |
|---|---|---|
| Incidence angle range | 35-45° | To allow for a wide swath a larger range in incidence angle was required (compared to space-borne configuration) |
| Frequency | 9.6 GHz (X-band) 17.2 GHz (Ku-band) | Proposed center frequencies for the CoReH2O mission |
| Polarization | VV + VH | Co- and cross-pol channels |
| Bandwidth | 100 MHz | |
| Spatial resolution | 10 x 10 m | with > 200 Equivalent Number of Looks (ENL). Data also available on request at 2 x 2 m spatial resolution. |
| Absolute bias | < 1 dB | |
| Inter-channel bias | < 1 dB | Di Leo et al. (2016), see section 2.2 |
| Radiometric stability | 95 % of data < 0.4-1.0 dB | Estimated from observations of repeated swaths, variable by site and channel. See section 2.2 |

## 2.2    Image processing

The processing of SnowSAR images is based on three steps:

1. Range compression
2. Doppler filtering
3. Ground back projection (GBP) algorithm

Overall gain of the processing chain is calculated based on the gain/loss contribution of each of these steps. Range compression introduces a gain from performing a non-normalized Fast Fourier Transform, being equal to the root square of the number of samples plus the windowing loss. Doppler filtering introduces further windowing loss, while the GBP algorithm includes gain from range dependency correction (equal to range distance to the power of four). The gain/loss of different processing steps is analysed in detail by DiLeo et al. (2016). This analysis shows that processing gain of SnowSAR is constant and predictable, and minimal inaccuracies are introduced during the focusing step of each pre-summed image.

## 2.3    Calibration and radiometric stability

Absolute calibration of SnowSAR backscattering intensity corrects for overall processing gain, from the acquisition of raw data to the generation of SAR images. Besides processor and receiver gains, nominal antenna gains and constant losses in the hardware are the main contributions addressed by calibration. In practice, calibration is based on a combination of three internal and external procedures. These are briefly described in the following sections.



### 2.3.1 Internal calibration

An internal calibration procedure defines the noise level and receiving gain of the SnowSAR instrument. The procedure is based on: 1) tracking of the transmitted power, and 2) estimating the noise level during every mission. Two dedicated power meters with 0.13 dB accuracy were used to measure the transmitted power of the two radar sub-systems (X and Ku frequency bands). Transmitted power was measured through a coupler with 30 dB attenuation for X-band and 27 dB for Ku-band. Measured power fluctuations (relative standard deviation) within a given mission typically were in the order of 0.1 dB or less for X-band and 0.25 dB for Ku-band. The Finnish missions, with a total of ten acquisitions over the site in Sodankylä, gave insight on stability of the mean transmitted power from one mission to another which varied by 1 dB for Ku-band and less than 0.5 dB for X-band (DiLeo et al., 2016).

Noise levels and receiving gain were estimated for each acquisition track of all SnowSAR missions directly from the range–Doppler map, during the processing of the images. In practice, a section of each flight track with assumed low noise was selected. In analyzing the noise variations, the X-VV channel noise was found to be unstable with large variations (several dB) apparent during a given mission. The noise of all other channels exhibited stable behavior, with noise level variation less than 0.5 dB during a single mission, and variations less than 1 dB from one mission to another. To compensate for the unstable behavior of the X-VV channel, a receiving gain equalization function was applied. A reference receiving gain level was extracted from the estimated gain over the entire campaign. Then, for each acquisition track a gain compensation factor for X-VV was calculated and applied.

### 2.3.2 External calibration

SnowSAR external calibration, calibrating the complete observed level of the detected backscatter, compensates for antenna patterns and receiver gain contributions. Corner reflectors with a known radar cross section (RCS) were deployed at each test site. Both trihedral and dihedral corner reflectors were deployed for calibration of co- and cross-polarization channels, respectively. However, due to the extremely narrow response of the latter in conjunction with the quite unstable motion of the small aircraft used, it was not possible to use the data of the dihedral corners, and only the co-polarized channels were calibrated by using the corners. The cross-polarized data were instead calibrated using a combination of comparative TerraSAR-X-imagery and tower-based radar observations (available for Finland only).

### 2.3.3 Radiometric stability

The radiometric stability, i.e. the capacity of the instrument to maintain a stable response to observed targets, was estimated by analyzing the calibrated SAR images from repeated tracks during a given mission, as well as by comparing sections of calibrated SAR images to the temporally closest TerraSAR-X observation available.





Calibration stability was found to vary depending on channel. There were also relatively large variations from one site to another. Table 2 summarizes the calibration stability in terms of differences between repeated tracks for flights conducted at each site, which indicate the level of maximum absolute difference (in dB) which covers 95% of the observations (overlapping grid cells).

**Table 2:** Maximum difference in SnowSAR $\sigma^0$ for calibrated images of repeated tracks for selected SnowSAR missions in Sodankylä,
Leutasch and TVC (from DiLeo et al., 2016)

| Site and mission | X-VV | X-VH | Ku-VH | Ku-VV |
|---|---|---|---|---|
| Sodankylä, NoSREx-M10 | 0.70 | 0.81 | 0.60 | 0.45 |
| Leutasch, AlpSAR-1 | 0.70 | 0.80 | 0.70 | 0.71 |
| TVC, TVCExp-2 | 0.62 | 0.56 | 0.35 | 0.40 |

The goal of 0.5 dB sensor stability was met for typically 85 % of all analyzed data. However, the difference between calibrated $\sigma^0$ on repeated tracks was less than 1 dB for 95 % of data. Further details, including a comparison with TerraSAR-X observations, are provided by Di Leo et al. (2016).

**3    Sites, collected SAR imagery and measured snow conditions**

**3.1    NoSREx campaigns in Finland – Sodankylä and Saariselkä**

The first flight using the SnowSAR instrument took place over a boreal forest site in Sodankylä, Finland on 17 March 2011. A total of ten science flights were flown at the site in the following winter between December 2011 and March 2012; additional two flights were performed in Saariselkä, Finland, representing tundra. These flights are collectively referred to as NoSREx
missions in this paper. Table 3 summarizes the dates of the NoSREx missions (labelled M00 to M10 for Sodankylä and T1 to T2 for Saariselkä), dates of the manual in situ measurements corresponding to each mission, which include notes in cases where exceptions were made to co-incident snow sampling and SnowSAR data acquisitions. The details of sites, flights and collected in situ data in Finland are briefly described in the following sections.

**Table 3:** Dates of NoSREx field campaigns and flight missions for the Sodankylä (Mxx) and Saariselkä (Txx) sites.

| Mission Id. | Field measurements | SnowSAR flights | Notes |
|---|---|---|---|
| NoSREx- M00 | 17.3.2011 | 17.3.2011 | 1st flight in winter 2010-2011. |



| | | | |
|---|---|---|---|
| NoSREx M01 | 20.12.2011 | 19.12.2011 | |
| NoSREx T01 | 20.12.2011 | 20.12.2011 | |
| NoSREx M02 | 9.1.2012<br>10.1.2012 | 9.1.2012<br>10.1.2012 | SnowSAR data acquisition spread over two days due to instrument malfunction on 9.1. |
| NoSREx M03 | 23.1.2012<br>24.1.2012 | 28.1.2012 | Flight delayed due to instrument malfunction, stable snow and weather conditions between flight and sampling. |
| NoSREx M04 | 7.-9.2.2012 | 7.2.2012 | Ground sampling extended due to stable snow and weather conditions. |
| NoSREx M05 | 22.-25.2.2012 | 22.2.2012 | Sampling was conducted daily between 5th and 6th missions, sampling can be considered equally applicable for both acquisitions. (no new snowfall). |
| NoSREx M06 | 26.2.2012 | 26.2.2012 | |
| NoSREx T02 | 29.2.2012 | 29.2.2012 | |
| NoSREx M07 | 1.3.2012 | 1.3.2012 | |
| NoSREx M08 | 5.3.2012 | 5.3.2012 | |
| NoSREx M09 | 8.3.2012<br>13.3.2012 | 10.3.2012 | Flight delayed due to adverse weather. Snow fall event of approximately 5 cm occurred between ground sampling and the flight (estimated based on AWS). |





| NoSREx M10 | 23.3.2012 | 24.3.2012 | Flight delayed due to adverse weather. Snow fall event of approximately 10 cm occurred between ground sampling and flight (estimated based on AWS). |
|---|---|---|---|

### 3.1.1 Sodankylä

Airborne measurements in Sodankylä covered an area of ca. 30 km$^2$ between the town of Sodankylä and Lake Orajärvi. This taiga landscape consisted mostly of forests of varying density on mineral soil, and open peatbogs (Figure 1). Flight transects were approximately 7 km in length. A total of 25 transects were planned in a general North-South orientation, and an additional two reference transects in a general East-West orientation. Measurements also covered some open fields and the westernmost end of the frozen Lake Orajärvi.

In situ data at Sodankylä included snow depth, density, and Snow Water Equivalent (SWE) measurements, snow pit observations at two fixed locations representing wetland and a forest opening, and hourly meteorological measurements. Snow depths along planned SnowSAR swaths were sampled at intervals of <10 m using the semi-automated MagnaProbe instrument (Sturm and Holmgren, 2018), or manually with a depth probe every 100 meters; SWE and snow density were recorded with a coring device every 500 meters (Hannula et al., 2016). Snow pit observations consisted of snow layering (stratigraphy), snow grain size, snow grain type, snow temperature profile and snow density profile (Leppänen et al., 2016).

At the Sodankylä site, not all flight transects could be sampled with in situ measurements. However, one parallel and one perpendicular transect were sampled during every campaign to provide a consistent reference. Observed snow conditions include wet/moist snow after initial accumulation, the dry snow accumulation period including snow metamorphism, and the effect of melt/refreeze events in late winter.

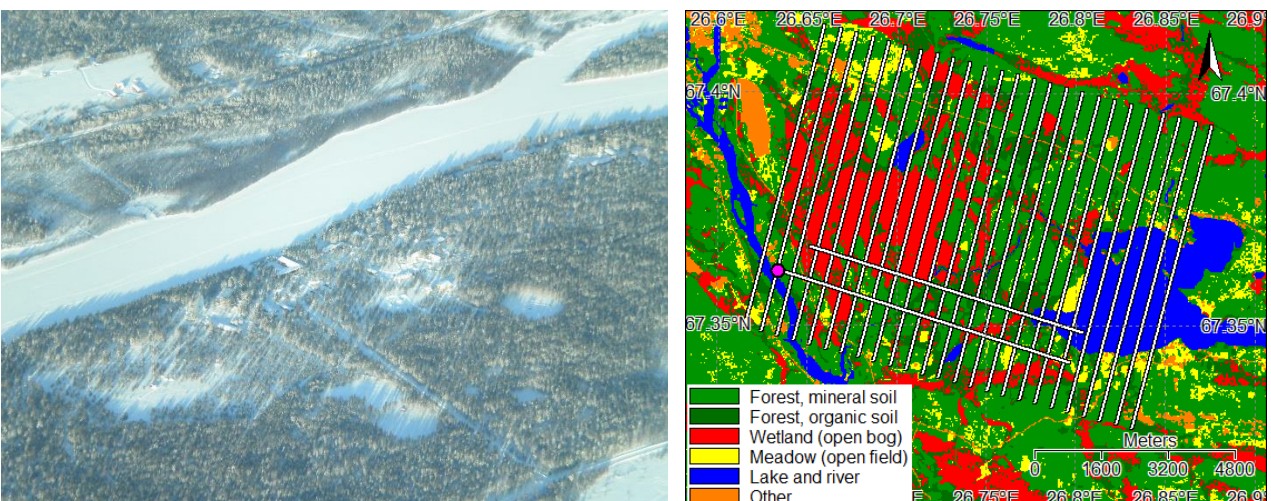

Figure 1: Aerial photograph (17 March 2011) and land cover map of the Sodankylä site with planned flight transects shown. Location of weather station indicated with purple dot. Photograph courtesy of T. Casal, ESA.

Ancillary data on DEM and vegetation height at 2 m resolution (based on LiDAR scans) provided tree height, canopy cover and stem volume data at 20 m spatial resolution (Natural Resources Institute Finland, https://www.maanmittauslaitos.fi/en/maps-and-spatial-data/expert-users/product-descriptions/laser-scanning-data). The high-resolution CORINE2012 database at 20 m resolution (https://www.syke.fi/projects/corine2012) was used as the basis for land cover classification to the generalized classes depicted in Figure 1 (see section 4.2 for details).


### 3.1.2   Saariselkä

Saariselkä, ca. 150 km north of Sodankylä (Figure 2), is a typical tundra landscape: barren rocky hills with very low vegetation cover and valleys with slightly denser low-lying vegetation but no significant forest cover. An automated weather station measuring snow depth, air and ground temperatures, and soil moisture profiles was located in the centre of the transect. A
single, ca. 20 km long SAR transect was flown in this area. Sampling of snow depth and SWE at the site was conducted similarly to Sodankylä. However, no snow pit observations were made at the Saariselkä site.

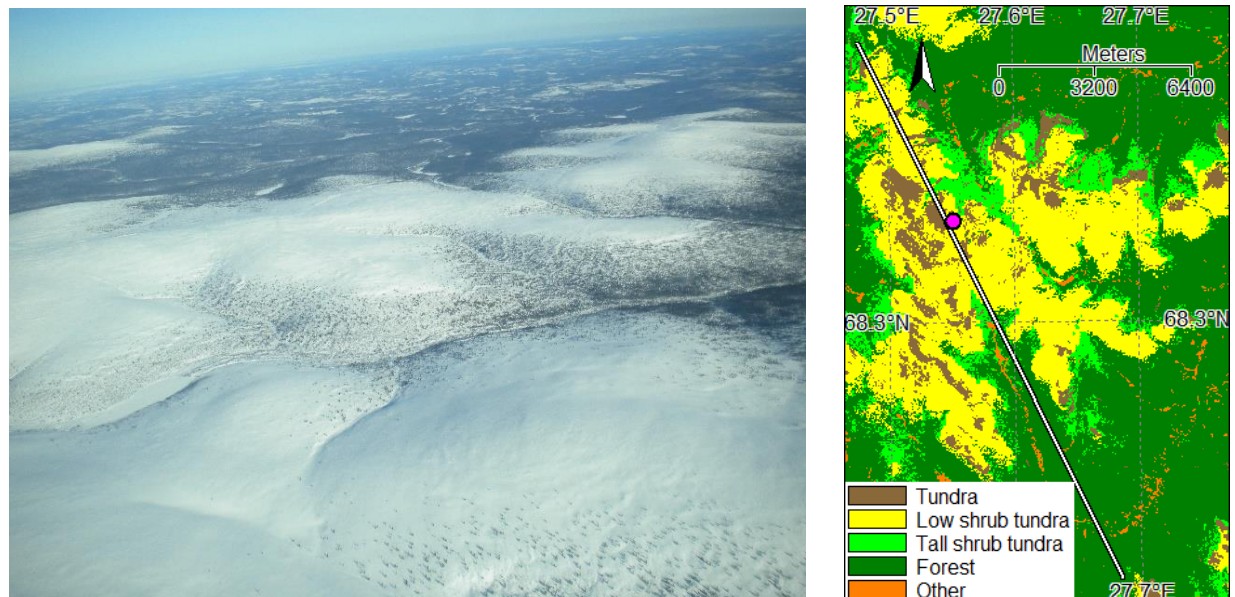

Figure 2: Aerial photograph (17 March 2011) and land cover map of Saariselkä with the single flight transect shown. Location of weather station indicated with purple dot. Photograph courtesy of T. Casal, ESA.

The two missions flown over Saariselkä cover initial accumulation of (dry) snow and then late winter conditions with wind-controlled redistribution and densification typical for tundra snow. DEM, LiDAR and land cover data from Saariselkä are identical to those available for Sodankylä.

### 3.1.3    SAR imagery

Airborne data acquisitions were planned to follow a 15-day repeat period between December and mid-February, corresponding to the repeat-pass time of the second phase of the proposed CoReH2O mission. Between February 22 and March 9, a three-day repeat period was planned, corresponding to the repeat pass time during the first phase of CoReH2O. A total of ten flights were flown at Sodankylä; as well as two flights over Saariselkä. Figure 3 shows example images of Ku-band VV backscatter collected over the Sodankylä and Saariselkä sites), exemplifying the variability of backscattering with land cover, snow conditions, and topography. For example in Sodankylä, relatively low backscatter (-10 to -12 dB) is observed over wetlands, while forested areas exhibit backscattering between -8 and -6 dB.



Figure 3: Examples of measured backscatter swath mosaics (in dB) over Sodankylä and Saariselkä, Finland during the NoSREx flights M05 (Sodankylä) and T2 (Saariselkä). Location of closest weather stations indicated by purple dots.


### 3.1.4    Meteorological and snow conditions

Figure 4 shows snow depths as well as air and soil temperatures measured by automated sensors at Sodankylä during the winters of 2010-2011 and 2011-2012. Snow depth information was available from acoustic depth sensors located in a forest

clearing, under the forest canopy and on an open wetland (bog) (Leppänen et al., 2018). All sensors exhibited similar responses to precipitation events during both winters, but the absolute magnitude of snow depths differed from one site to another, with the sensor in the forest clearing exhibiting the highest depths.

The early winter season of 2010-2011 (Figure 4, upper panel) was characterized by persistent shallow snow cover (< 20 cm in the clearing) combined with temperatures remaining below freezing from 10 November onwards. This resulted in a rapid

freezing of the soil, due to relatively low insulation from the shallow snowpack. Snow depth increased gradually, with several major precipitation events of over 10 cm from January to March. A maximum snow depth of 80 cm (ca. 170 mm in SWE) in the forest clearing, was measured in March.

Before the first SnowSAR flight on 17 March 2011 (M00), several small melt/refreeze events occurred on 3, 6 and 8 March 2011, causing densification of layers near the surface of the snowpack. No melt events were recorded during the SnowSAR

flight. Measured air temperatures ranged between -14 °C at night to 2 °C during daytime (on 3 March 2011) for 1-7 March 2011 and between -8 °C and -1°C for 17 March 2011.

Soil probes indicated a rapid freezing of soil surface layers after mid-November. Air temperatures exceeding 0°C in early March did not increase measured soil moisture (not shown), suggesting that only light surface snow melt had occurred. Soil temperature, measured at a depth of 2 cm, as well as the soil moisture sensors (not shown), indicated the onset of strong snow

melt only after 10 April 2011.

During the winter of 2011-2012 (Figure 4, lower panel) the total snow depth increased from 10-30 cm during M01, to 50-80 cm during M10. The largest increase in snow depth (approximately 14-20 cm) between two consecutive SnowSAR missions occurred between M04 and M05. Soil temperature remained close to 0°C until late January 2012 due to the insulating effect of the relatively deep snowpack, indicating residual unfrozen water in the soil. This was also apparent in direct measurements

of soil permittivity (Lemmetyinen et al., 2018).

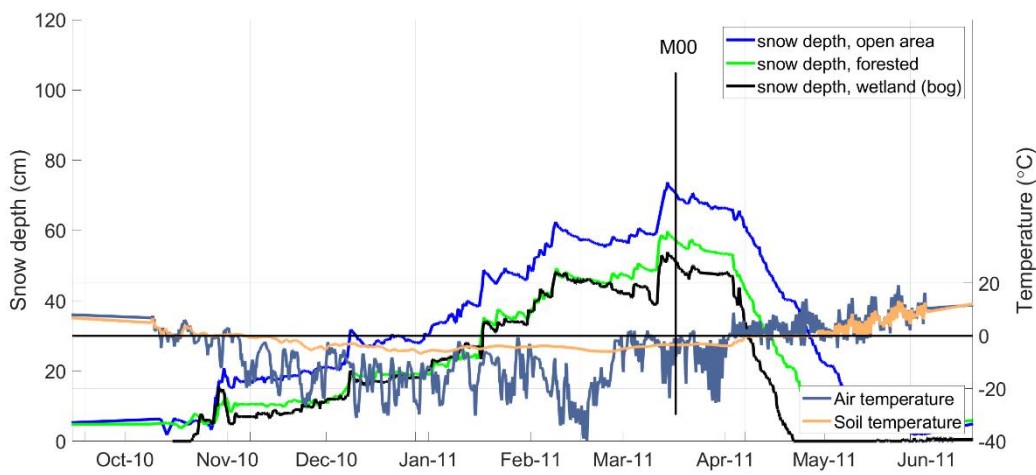





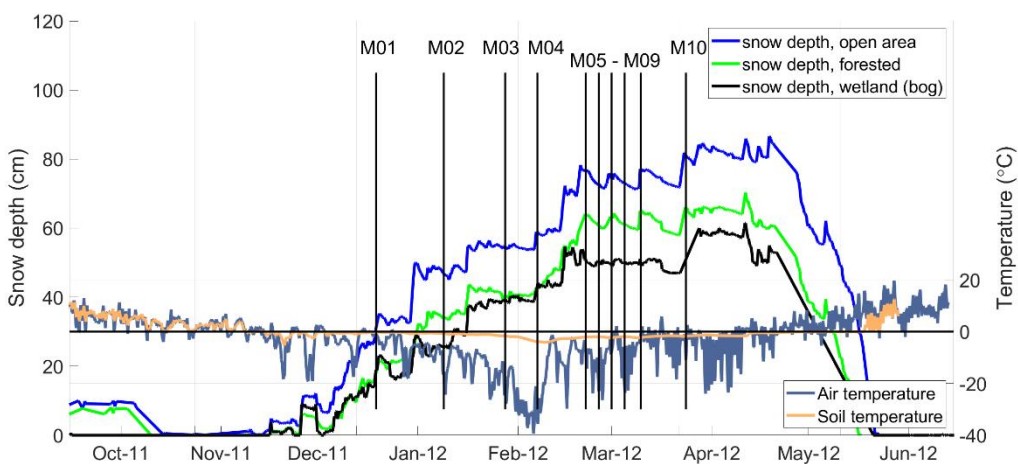

Figure 4: Snow depth, air and 2 cm soil temperature at Sodankylä during 2010-2011 (upper panel) and 2011-2012 (lower panel) campaigns. Timing of the SnowSAR acquisitions (M00-M10) indicated with vertical lines.

Snow depth, air and soil temperatures at Saariselkä are presented in Figure 5. Due to frequent high winds, the snow depth distribution is highly variable, so measured depth at a single site is only an approximate measure of snow accumulation during the season. Similar to Sodankylä, soil temperatures during the first SnowSAR mission T1 were still close to 0°C, but the soil was frozen by T2.

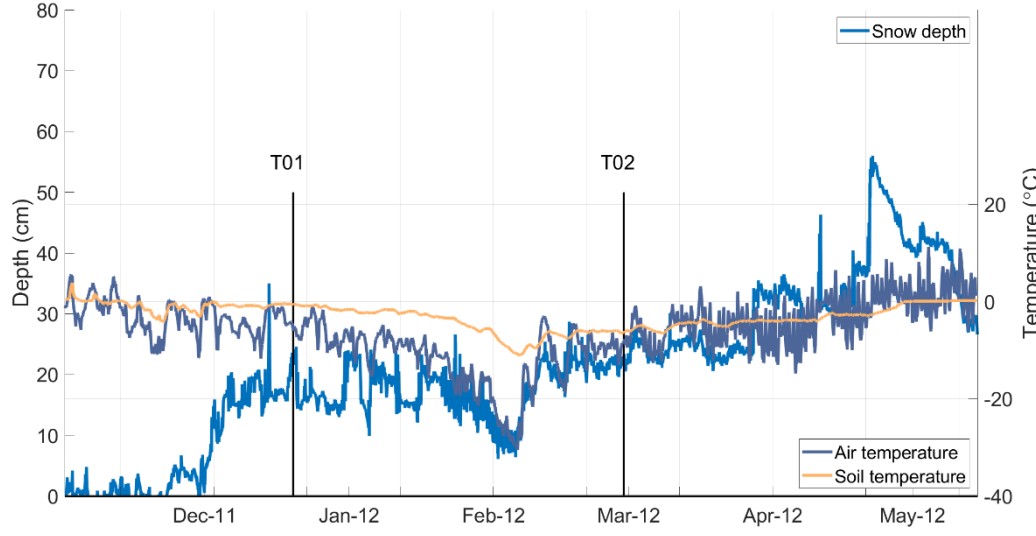



Figure 5: Snow depth, air temperature and 5 cm soil temperature at Saariselkä during the winter of 2011-2012. Soil temperature from a mineral soil site at a depth of 5 cm. Timing of two SnowSAR acquisitions (T01 and T02) indicated with vertical lines.


The manual snow survey program of the Finnish Meteorological Institute (FMI) measured snowpack evolution over two predominant land cover types in the Sodankylä region (boreal forest and wetlands; Leppänen et al., 2016). During the NoSREx flight campaigns, snow pit measurements were made coincidently with SnowSAR flight dates. Examples of manually observed snow profiles at the forest clearing site are shown in Figure 6 for 17 March 2011 (M00) and several dates during the winter of

2011-2012. The snowpack showed a complex vertical structure during both winters, typical of the boreal forest region. The typical observed grain size in March 2011 (Figure 6a), from macro-photography of snow samples, reached 2.5-3 mm in depth hoar layers in the bottom of the snowpack. In the winter of 2011-2012, most observed typical grain size values were 1 to 1.5 mm throughout each winter. During M01 (Figure 6b), the lower layers of the snowpack were at 0°C, indicating the presence of liquid water, which was likely to affect observed backscatter for that date.


Figure 6: Examples of measured snow profiles in a forest clearing site during NoSREx missions: (a) M00 (17 March 2011)
and in the winter of 2011-2012, b: M01 19 December 2011; c: M02 January 9, 2012; d: M04 February 8, 2012; e: M09 March
8, 2012. Profiles include density (green line), temperature (red line), grain size (minimum-maximum ranges), grain type and
hand hardness (vertical bars; K=knife, P=pencil, 1F=one finger, 4F=four fingers and F = fist). Hand hardness was not estimated
on 17 March 2011.

Average snow conditions (snow depth, density and SWE) measured by in situ transects are shown in Table 4. The number and
location of flight transects varied per mission, which may result in uncertainties when comparing one mission to another, as



exemplified by the apparent decrease in snow depth between M09 and M10. The average depth-weighted grain size from the two regular snow pits is also shown.

Table 4: Snow depth (SD), snow density and SWE measured from transects during NoSREx Sodankylä missions M00-M10 and Saariselkä missions T1 and T2.

| Mission ID. | SD samples (#) | SD mean, std (cm) | SWE samples (#) | Snow density (kg/m³) Mean, std | SWE (mm) Mean, std |
|---|---|---|---|---|---|
| M00 | 213 | 57, 13 | 42 | 205, 44 | 114, 24 |
| M01 | 587 | 26, 6 | 26 | 193, 87 | 44, 9 |
| M02 | 718 | 34, 11 | 40 | 231, 75 | 69, 25 |
| M03 | 1322 | 41, 14 | 70 | 190, 29 | 79, 26 |
| M04 | 1231 | 45, 16 | 70 | 193, 26 | 92, 33 |
| M05 | 9824 | 60, 13 | 118 | 200, 22 | 124, 27 |
| M06 | 703 | 60, 13 | 47 | 203, 14 | 122, 19 |
| M07 | 2940 | 63, 11 | 54 | 210, 18 | 132, 26 |
| M08 | 117 | 56, 14 | 25 | 231, 25 | 130, 32 |
| M09 | 500 | 59, 13 | 34 | 232, 20 | 135, 30 |
| M10 | 208 | 52, 23 | 36 | 222, 22 | 133, 45 |
| T1 | 16 | 35, 16 | 8 | 202, 32 | 59, 30 |
| T2 | 3783 | 51, 28 | 71 | 274, 42 | 127, 94 |


## 3.2 Austrian Alps – AlpSAR campaign

The AlpSAR 2012/13 campaign took place over three sites located in different elevation zones of the Austrian Alps. Each individual site is of limited spatial extent as the flight tracks extend along specific orographic features, i.e. narrow valleys for

Leutasch and Rotmoos, respectively, and a glacier (Mittelbergferner). Consequently, differences in snowpack properties between the sites are much larger than the variability within an individual site. This is also the case for the temporal variability of processes affecting the snow metamorphic state, where at the lowest site (Leutasch, elevation ca. 1150 m a.s.l) several melt/freeze events happened during the period between the three flight campaigns, whereas no melt event occurred at the two other sites (Rotmoos, ca. 2300 m a.s.l. and Mittelbergferner, 2500 to 3350 m a.s.l.) The temporal evolution of parameters of

the snowpack and upper soil layers were recorded at Leutasch and Rotmoos. Time series of meteorological parameters are also




available from these special stations, as well as from nearby operational automatic stations of the Austrian Meteorological Service , including a station at 2840 m elevation, 1 km from the Mittelbergferner glacier. Spatial variability in snowpack properties was measured during field campaigns coincident with the three flight missions (Table 5). An additional field campaign was conducted during the 2nd week of January 2013, scheduled to coincide with the 2nd flight mission that was shifted due to problems with flight conditions.

**Table 5:** Dates of the field campaigns and flight missions for AlpSAR sites.

| Misson ID | Field measurements | Date of SnowSAR flights |
|---|---|---|
| AlpSAR-1 | 19. -25.11.2012 | 21.11.2012 |
| AlpSAR-2A | 8.-14.1.2013 | No flight; comprehensive field measurements |
| AlpSAR-2B | 21.-25.1.2013 | 24.1.2013; reduced field data collection |
| AlpSAR-3 | 17.-26.2.2013 | 21. and 23.2.2013 |

### 3.2.1 Leutasch

The Leutasch valley (47.36°N, 11.14°E) extends from south-west towards north-east between the steep Wetterstein mountain range in the north and the densely forested hills of the Seefeld/Wildmoos plateau to the south. In the main data acquisition area, surface height decreases from 1158 m to 1100 m a.s.l. over a horizontal distance of 5 km. Snowfall events can be intense due to orographic effects, especially when frontal systems approach from the north-west. During the winter season a network of cross-country skiing tracks runs through the valley, which are visible in backscatter images as backscatter intensity is reduced due to compaction.

The land cover map (Figure 7, right) is based on the Digital Air Photo and Lidar Atlas, Geo-information Division, Government of the Province of Tyrol. The main land cover class in the level part of the valley is meadow/fields, dominated by cultivated meadows which were snow-free and in dormant stage during the 1st campaign in November 2012. The main forest type is dense coniferous forest, which dominates the slopes, but is limited in extent in the valley floor. Electricity pylons cover a very small part of the area, but are included in the map because their metal frames have high radar reflectivity.



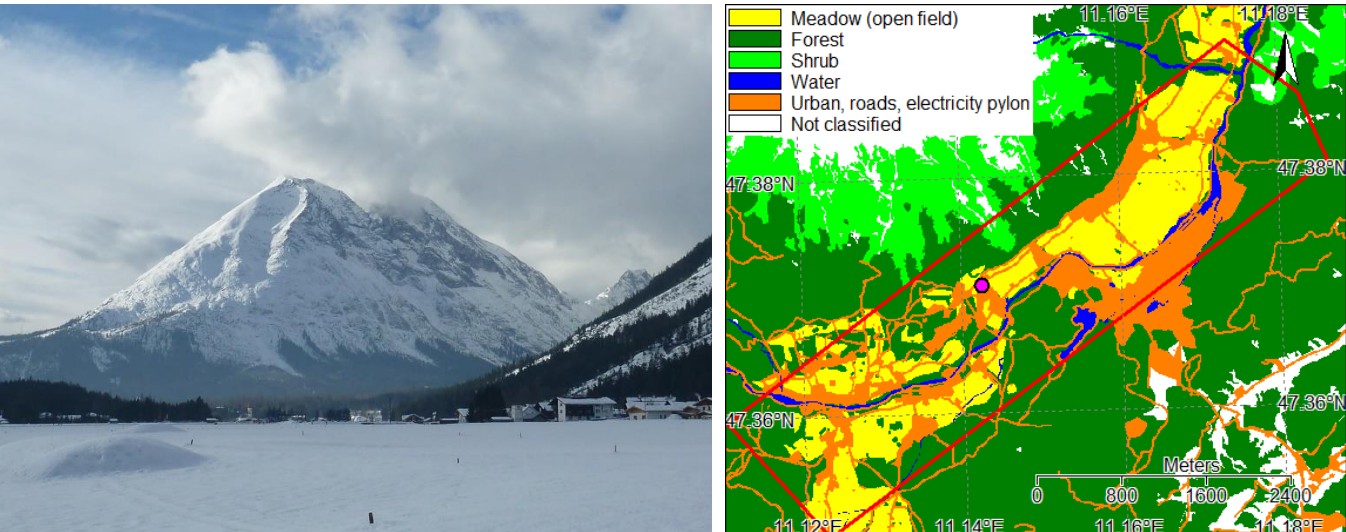

Figure 7: Photograph (10 January 2013) and land cover map of Leutasch with approximate area covered by SnowSAR indicated within the red line. Location of weather station indicated by purple dot.


### 3.2.2 Rotmoos

The Rotmoos valley (46.82°N, 11.06°E, **Figure 8**) is a small tributary to the Gurgler Tal and the Ötztaler Ache (river). The surrounding topography includes glacier-covered mountain ridges in the east, south and west, reaching the highest elevation (3426 m) in the south-west corner. Snow depth and physical properties were measured in a narrow central section of the valley
where the surface rises NW to SE in direction, between 2250 m to 2400 m in elevation.

The Rotmoos valley is a long-term international biological and ecological monitoring site, supported by the Alpine Research Centre Obergurgl of the University of Innsbruck (Koch & Erschbamer, 2010). Detailed maps on vegetation cover and geology, have been aggregated into five classes (**Figure 8**, right): alpine grassland, sparse alpine vegetation, bog/marsh,
rock/scree/moraine and alluvium. These classes are based on the Biotope Map for Protected Areas, Digital Air Photo and Lidar Atlas, Geo-information Division, Government of Tyrol and the normalized difference vegetation index derived from top-of-atmosphere reflectance (corrected for topographic effects and atmosphere) in the ETM bands 4 (775 nm – 900 nm) and 3 (630 nm – 690 nm) of a Landsat image of 21 Aug. 2011.


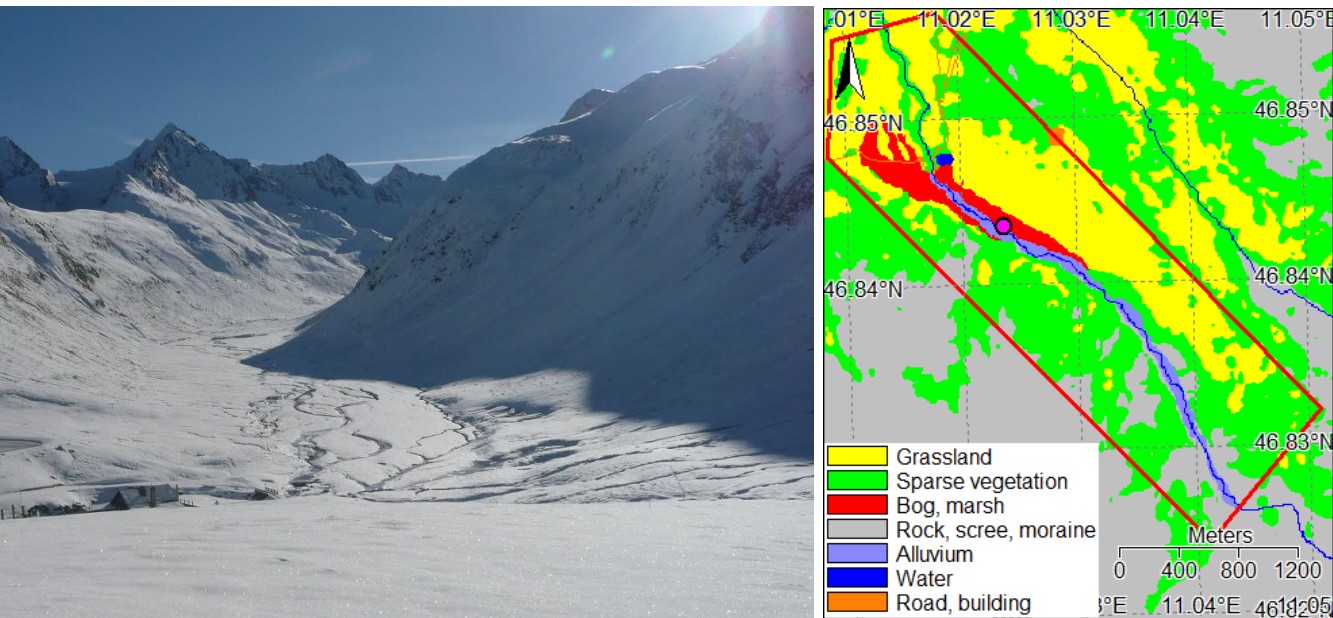

**Figure 8:** Photograph (20 November 2012) and land cover map of the Rotmoos valley with approximate area covered by SnowSAR indicated within the red line. Location of weather station indicated by purple dot.

The upper part of the Rotmoos site is morphologically characterized by glacier foreland, the lower part and the south-west facing slopes are mainly covered by Alpine grassland (sedges, grasses, and locally dwarf shrubs). North-east facing slopes are

steeper, covered by sparse Alpine vegetation (patchy cover of sedges). A substantial part of the level section is covered by bog, including the site of the automated meteorological and snow station and of the ground-based SAR measurements (Rekioua et al., 2017). The river bed along the valley floor is up to 100 m wide and covered by coarse material. On the orographic right side (east side) of the valley, about 1 km above the confluence with the Gurgler Ache, is a reservoir supplying water for snow production to the adjoining skiing area Obergurgl. The Rotmoos valley is a nature reserve, not affected by ski pistes. During

the campaigns, the snow along the flight tracks was undisturbed.

### 3.2.3 Mittelbergferner

Mittelbergferner (46.92°N, 10.89°E, Figure 9) is a northerly exposed glacier in the Ötztal Alps, covering about 9 km² in area over the elevation range from 2500 m to 3552 m a.s.l. Field measurements were performed on the main branch of the glacier

at elevations between 2700 and 3200 m. Due to warm summers during the previous two decades, the firn area was confined to north facing slopes above about 3100 m elevation. The ice area (corresponding to exposed ice surfaces in late summer) dominates in extent over the firn area, a clear indication of negative mass balance for many years. The AlpSAR project





addressed the feasibility for measuring the accumulation of seasonal snow on a glacier. At X- and Ku-band frequencies the backscatter signal of the glacier during the cold period is largely dominated by the contributions of the refrozen snow and firn
below the seasonal snow cover. Consequently, the state of the medium below the seasonal snow has a large impact on the backscatter behaviour. The pattern of snow accumulation during winter depends on small scale orographic effects during snowfall and on the redistribution of deposited snow due to wind. It shows a general trend of snow depth increase with elevation.

The map of surface cover (Figure 9, right) is derived from a geocoded, cloud-free Landsat image acquired on 21 August 2011. For discriminating the total glacier area versus ice-free surfaces (rock, rubble, moraine), the normalized difference snow index was used, based on ETM band 5 (1550 nm – 1750 nm) and band 3. For discriminating ice and firn areas, a threshold in the top-of-atmosphere spectral reflectance image (corrected for topographic effects and atmosphere) in ETM band 4 was applied. Firn (metamorphic snow from previous years) and surfaces with remnant refrozen snow from the last winter were merged into
one class (firn area), because the albedo does not allow a clear discrimination of these two sub-classes. The snow from the 2011/12 winter, not melted out during summer 2012, rested on firn, so that the radar signal intensity as background to the 2012/13 winter snow is similar for both sub-classes. While there was no suitable cloudless Landsat image available in late summer 2012, an oblique aerial photo of 11 September 2012 indicated little change in the extent of glacier ice and firn areas between the two years. Measurements on firn structure at the firn temperature site (3110 m a.s.l.) show ice layers with large
air bubbles between layers of compacted coarse-grained snow, being efficient scattering elements. This site is located slightly above the upper boundary of the ice area, leading to the conclusion that the top firn layers were deposited as snow several years ago and had since been subject to several melt/freeze cycles.

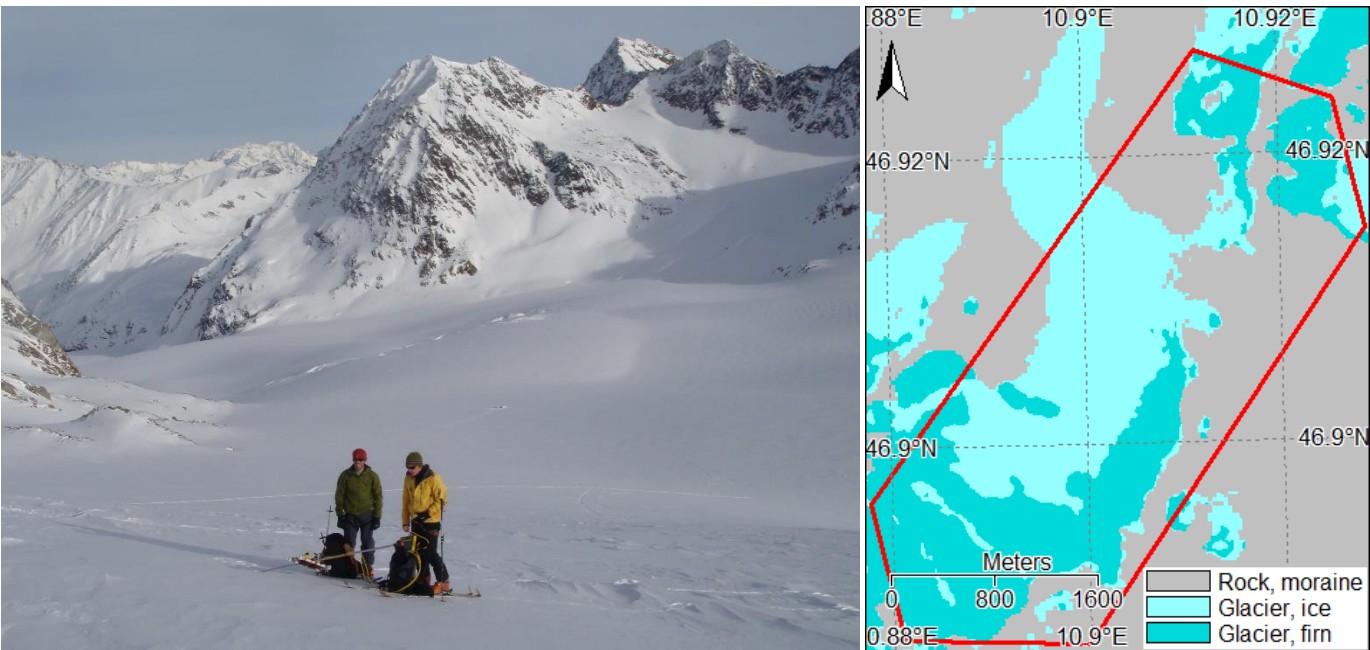

Figure 9: Photograph (9 January 2013) and surface cover map of Mittelbergferner with approximate area covered by SnowSAR indicated within the red line.

### 3.2.4 SAR imagery

Figure 10 gives an example of SAR data collected over the AlpSAR test sites of Leutasch, Rotmoos and Mittelbergferner. Ku-
band VV-pol collected on 21 and 23 February 2013 are shown. Local topography influences measurements in particular over Rotmoos and Mittelbergferner, and gaps apparent within individual tracks are mainly attributed to steep slopes. The highest backscatter values (-3dB to -5 dB) were observed over the open field sites at Leutasch due to the dominant backscatter of refrozen snow. The low backscatter values at Rotmoos refer to back-slopes, but also at 40 degree angles the backscatter intensities are lower by several dB compared to Leutasch. On Mittelbergferner the average backscatter intensities are
comparable to those of Rotmoos. Slightly higher values are observed in the firn area (in the south-east sector).

Earth System
Open Access Science
Data Discussions

Figure 10: Examples of collected SAR imagery over Mittelbergferner, Rotmoos and Leutasch during AlpSAR-3. KuVV depicted. Purple dots indicate locations of recording snow and weather stations at Rotmoos and Leutasch.


### 3.2.5    Meteorological and Snow Conditions

Stratigraphy and physical properties of snow (grain size and type, density profile, temperature profile) were measured in snow pits; snow depth was measured along transects using conventional techniques (depth probe) at regular intervals. Other
techniques, such as the Snow-MicroPen (Proksch et al., 2015) were applied to study snow characteristics but are not included in the present dataset. At Rotmoos, two experimental devices were used to map changes in snow depth: a terrestrial laser scanner and digital stereo-photography acquired from a RPAS (Remotely Piloted Aircraft System) (Rott et al., 2013); however these data are not included here. Here we show examples of snow pit measurements and of recorded atmosphere, snow and





soil parameters. Mean values of snow depth, density and SWE collected in each of the three sites are listed in Table 6. The
mean densities of the snow pits at a given site and date show only small differences. Mean SWE is computed from the mean
SD of all points using the mean density of the pits.

Table 6: Mean and standard deviation (std.) of snow depth (SD) from depth transects. The mean density is from snow pit
measurements. Mean SWE refers to the mean SD of the transects multiplied by the mean density from the snow pits. The data
for Mittelbergferner refer to seasonal snow accumulating after 1 October 2012.

| | Leutasch | | | | Rotmoos | | | | Mittelbergferner | | | |
|---|---|---|---|---|---|---|---|---|---|---|---|---|
| Mission Id. | SD samples (#) | SD (cm) Mean, std. | Density (kg m$^{-3}$) | SWE (mm) | SD samples (#) | SD (cm) Mean, std. | Density (kg m$^{-3}$) | SWE (mm) | SD samples (#) | SD (cm) Mean, std. | Density (kg m$^{-3}$) | SWE (mm) |
| AlpSAR-1 | | - | - | - | 190 | 27, 9 | 333 | 90 | 360 | 85, 14 | 307 | 261 |
| AlpSAR-2B | 185 | 48, 9 | 315 | 154 | 120 | 126, 19 | 264 | 333 | 100 | 179, 21 | 314 | 562 |
| AlpSAR-3 | 216 | 75, 7 | 301 | 226 | 805 | 129, 22 | 306 | 395 | 354 | 215, 28 | 328 | 705 |


**Leutasch**

During the 1st flight campaign in Leutasch on 21 November 2012 the surface was snow-free. Accumulation of winter snow
cover started on 28 November 2012, with the main build-up of the snowpack during a cold period before mid-December
(**Figure 11**). Between mid-December 2012 and 7 January 2013 there were three melt events, the last one with substantial
rainfall, which caused wetting down to the base of the snowpack. During field campaign 2A (snow pits on 10 January 2013)
the lower part of the snowpack was still wet, with a solid frozen crust of 10 cm thickness on top. Between 10 January and the
2nd flight mission (24 January) the complete snowpack refroze and some fresh snow accumulated on top (field campaign 2B).
The flights on 24 January and 21 February 2013 took place during cold periods when the snowpack was completely dry. Due
to transient melt events before flight Mission 2 the lower 40 cm of the snowpack were composed of coarse-grained refrozen
snow (grain size ranging from 1.0 to 2.5 mm) with some thin ice layers (**Figure 12**). The coarse-grained layer was covered by
about 10 cm of fine-grained snow during Mission 2 and by about 35 cm of fine-grained snow during Mission 3. Between
Missions 2B and 3 about 75 mm SWE accumulated on the average, totalling 226 mm of SWE and a snow depth of 75 cm on
23 February 2013. The structure of the lower 40 cm of the snowpack during Mission 3 was similar to Mission 2 (coarse
grained). The snowpack above the refrozen layer was made up by several thin layers arising from sequential snowfall and short
melt events, with a thin top layer of fine-grained fresh snow. Snow layers above the refrozen crust were made up by rounded



grains and solid faceted crystals, with a grain size between 0.2 mm and 1.0 mm. Soil temperatures at depths of 3 cm and 7 cm remained above 0°C throughout the snow cover period.

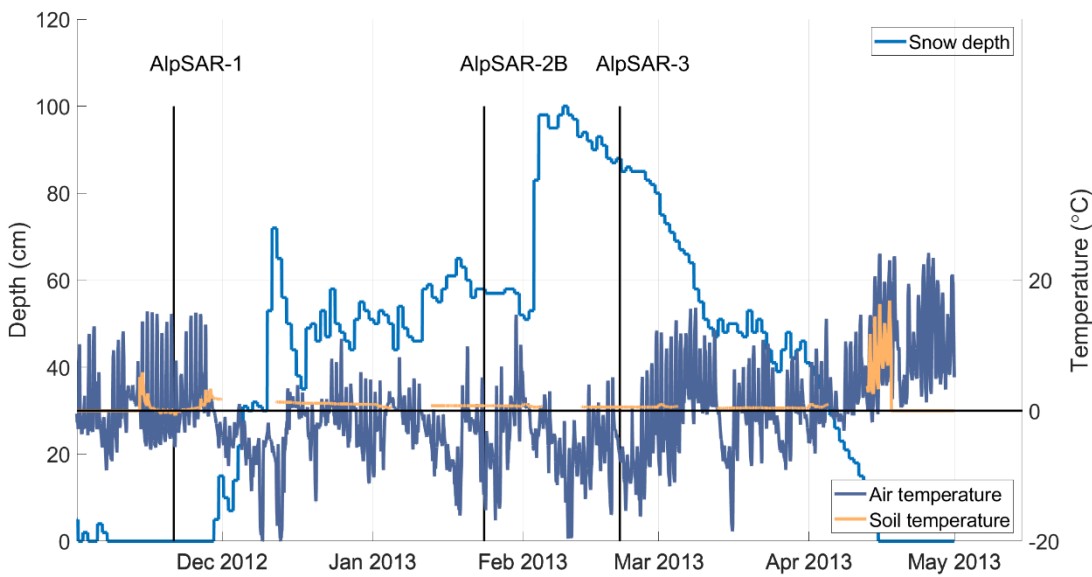

**Figure 11.** Time series of air temperature, soil temperature (3 cm depth) and snow depth at AWS Leutasch (11.1425 deg. E., 47.3708 deg. N, 1135 m a.s.l.). The vertical lines indicate the dates of the three flight campaigns.





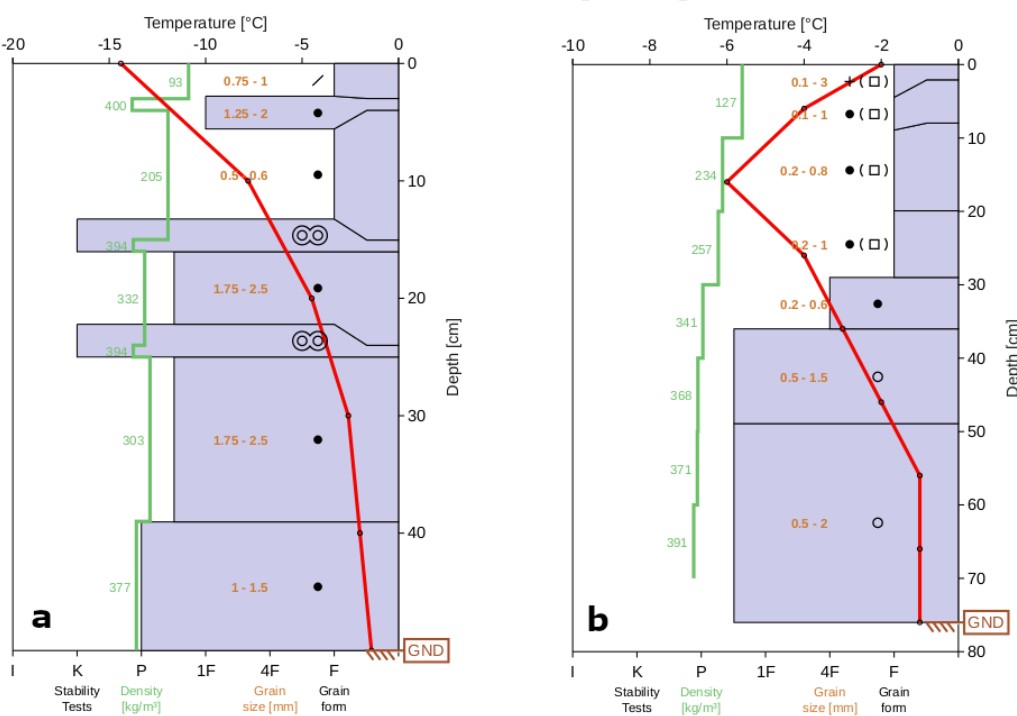

**Figure 12.** Examples of snowpack profiles measured in Leutasch, AlpSAR campaigns 2B (a) and 3 (b) from snow pit 3.


### Rotmoos

An automated station operated in the bog area at 2265 m elevation between 24 October 2012 and 27 June 2013, measuring the following variables at 5-minute time steps: air temperature, humidity, pressure, wind speed and direction, shortwave solar irradiance; snow properties: depth, temperature (20cm, 40cm, 60cm, 80cm, 100cm, 120cm, 140cm, above ground); soil
properties: temperature and wetness (3 cm, 11cm, 30 cm below the surface). Snow depths, air and soil temperatures are presented, as an example, in **Figure 13**.

The first snowfall event was on 27-28 October 2012, building up a snow layer of 35 cm depth. The soil below the snowpack remained unfrozen throughout winter except for the top 2 cm. There were two short melt-freeze events during the first two
weeks of November. Between mid-November 2012 and mid-April 2013 no significant melt event occurred. The snowpack was completely dry during the flight missions. During the first campaign on 20-21 November the ground was covered by 20 to 35 cm of re-frozen coarse-grained snow. The mean snow depth along the transects was 27 cm, and mean SWE was 90 mm. The coarse-grained, refrozen layer of about 30 cm depth from November stayed throughout winter, with some formation of depth hoar (especially at the bog sites). Between the flight missions M1 and M2 there was one major snowfall event of about
40 cm on 28 November 2012, followed by several small events. Average snow depth increased to 126 cm, and mean SWE

(based on the mean density of the snow pits) to 333 mm. Between the flight missions M2 and M3 there were again some minor snowfall events. The mean snow depth increased only slightly to 129 cm whereas SWE increased to 395 mm, indicating snow compaction. During M2 and M3, snow layers above the refrozen bottom layer included solid faceted particles and also rounded grains (**Figure 14**). Snow depths along the transects showed small-scale spatial variability related to micro-topography and

snow redistribution by wind.

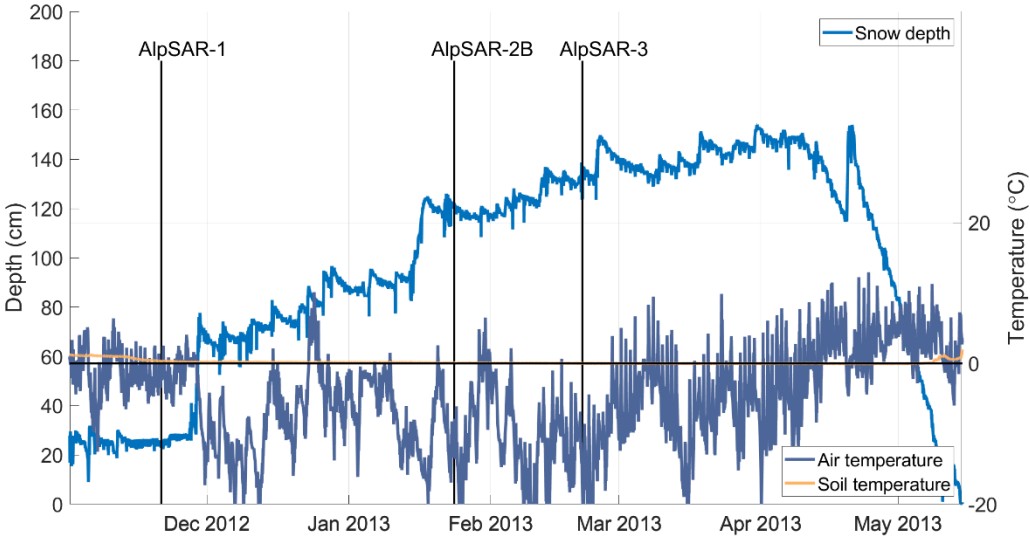

**Figure 13.** Time series of air temperature, snow depth and soil temperature (3 cm depth) at AWS Rotmoos (11.0236 deg. E, 46.8435 deg. N, 2266 m a.s.l.). Vertical lines indicate the dates of the SnowSAR flight missions.






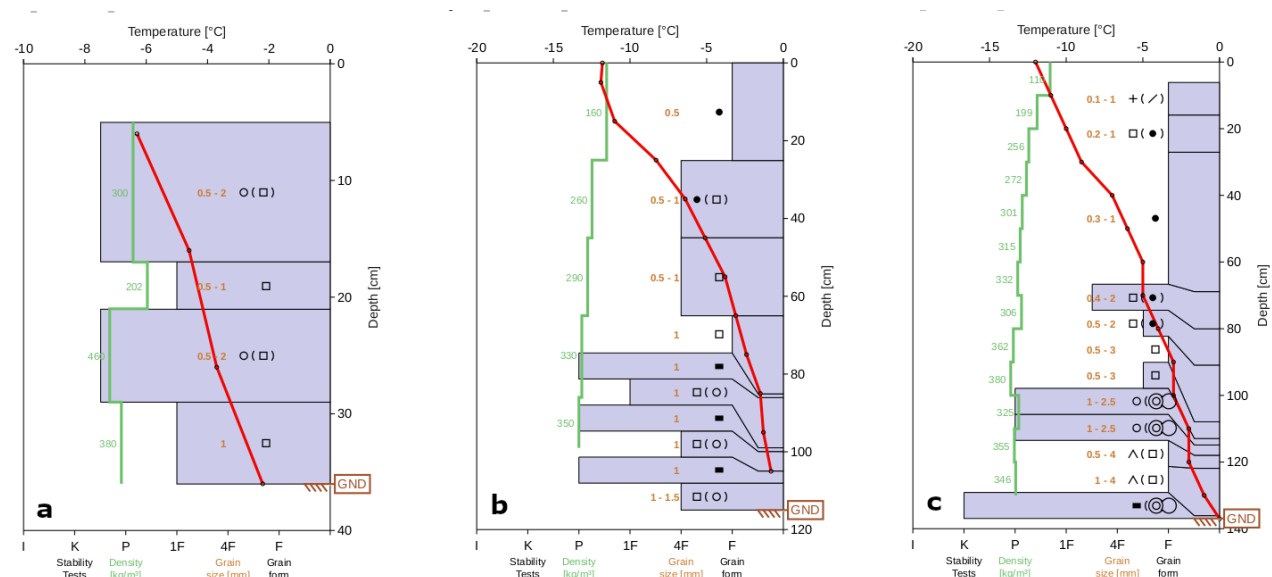

**Figure 14**. Examples of snowpack profiles measured at Rotmoos, AlpSAR campaigns 1 and 2B from snow pit 3 (a-b) and campaign 3 from snowpit GB-SAR (near snow pit 3) (c).


### Mittelbergferner

Measurements at Mittelbergferner were concerned with the feasibility of measuring seasonal snow accumulation on a temperate glacier. Snow depths and snow pit data refer to the seasonal snowpack above glacier ice or frozen firn. Measurements were madeon the glacier at elevations between 2700 m to 3200 m a.s.l. Up to elevations of about 3100 m the background

medium below the winter snowpack was mainly glacier ice, and above about 3100 m the background medium was frozen firn composed of coarse grained, clustered melt-metamorphic snow with ice layers of several cm thickness. Main snowfall events after the summer melt period happened between mid-October and mid-November 2012. The snow of October was subject to melt-freeze cycles, resulting in a coarse-grained bottom layer of about 30 cm thickness that persisted during winter. During Mission 1 the glacier ice and firn were already frozen down to several metres depth and were covered by a seasonal snow

cover of 85 cm mean depth (mean SWE of 261 mm).

The structure and morphology of the seasonal snowpack did not change much between M1 and M3. Above the coarse-grained metamorphic base layer, several layers with fine to medium sized grains accumulated, resulting in a mean snow depth of 2.15 m and SWE of 705 mm during M3. Due to compaction, the density increased slightly and temperature gradient metamorphism

caused a minor increase in grain size through formation of faceted crystals (Figure 15). Typical grain size of the snowpack above the basal layer ranged from 0.5 mm to 1.0 mm. Because the X- and Ku-band backscatter signal was dominated by the




contributions of the frozen glacier ice, firn and bottom layer of refrozen snow, accumulation of seasonal snow with a smaller scattering albedo, was associated with a slight decrease in the total backscatter coefficient.

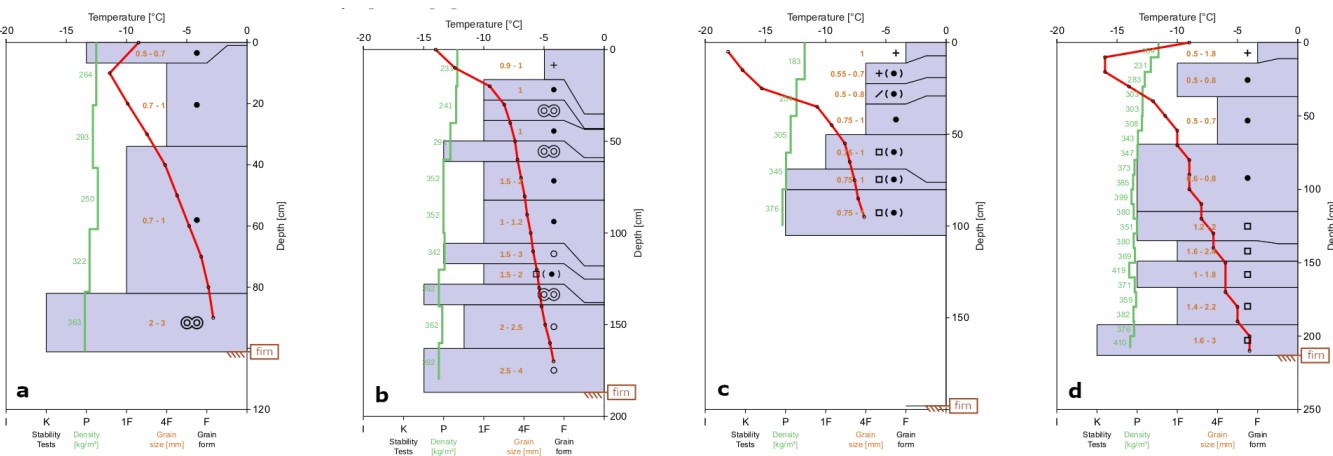


Figure 15. Examples of snowpack profiles measured on Mittelbergferner, AlpSAR campaigns 1, 2A, 2B and 3 (a-d) from snow pit 4. Note that the full profile was not measured during campaign 2B (panel c) due to lack of time. However, for the lower part of the pack, snow properties are expected to have changed little since measurements from campaign 2A (panel b).

**3.3    Trail Valley Creek, Canada**

Trail Valley Creek (TVC) is a 58 km$^2$ research basin established in 1991 north of Inuvik, Northwest Territories, Canada (68°45'N, 133°39'W;


Figure 16). Situated near the northern edge of the boreal forest, land cover is predominantly tundra-like with isolated shrub and forest patches. Thawing of the once continuous permafrost, rapid expansion of shrub vegetation, and earlier snowmelt have contributed to substantial land cover change and ecological impact over the last 30 years (Wilcox et. al., 2019, Grünberg et. al., 2020). Shallow snow commonly observed within the basin (<0.3 m) is typical of a tundra environment, however deep

drifts (>2 m) form in proximity to tall standing vegetation and steep terrain (Essery et al., 2004, King et al., 2018).



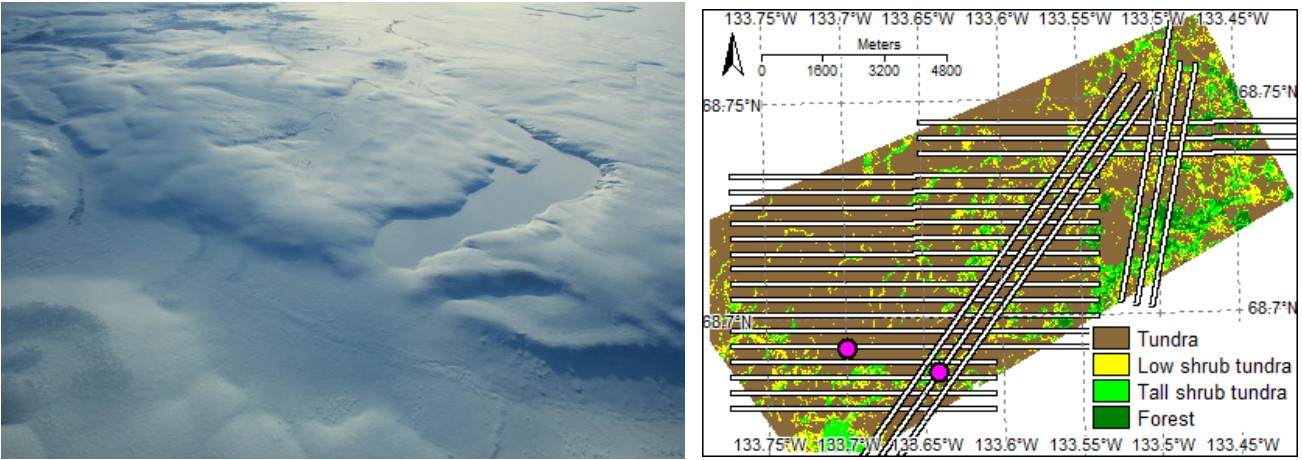

Figure 16: Aerial photograph (left) and land cover map of the Trail Valley Creek (TVC) research basin, overlaid by SnowSAR
flight transects. Location of weather stations indicated by purple dots. Photograph courtesy of A. Coccia, Metasensing.

Two SnowSAR flights were performed in March and April 2013. The flight in December 2012 was cancelled; however, collection of in situ data took place (Table 7).


**Table 7:** Dates of the field campaigns and flight missions at TVC.

| Mission Id. | Field Measurements | SnowSAR Flights |
|---|---|---|
| TVCEx-1 | 11.-15.12.2012 | No flight |
| TVCEx-2 | 8.-17.3.2013 | 13.-14.3.2013 |
| TVCEx-3 | 5.-9.4.2013 | 8.-9.4.2013 |

The snow sampling strategy at TVC was devised to determine:

1. variability in bulk snow properties (depth; density; SWE) within and between landscape units;
2. variability in snow stratigraphic properties (layer density; grain size) within landscape units and down to the resolution of airborne radar re-sampling (~5 to 50 m);
3. SWE stored in large drift features via ground-based LiDAR surveys (data not provided as part of SnowSAR dataset)





Transects of georeferenced snow depths with approximately 3 m spacing, were measured using Magnaprobes throughout the research basin. More broadly spaced (~200 - 500 m) ESC-30 snow core measurements provided bulk snow density measurements needed to convert magnaprobe snow depths to SWE. Snow stratigraphic measurements were made using conventional snowpit observations (layer identification; density profiles with 100 cc cutters; visual grain dimension estimates) and objective techniques of snow specific surface area (SSA) following Gallet et al. 2009. In order to understand how

stratigraphic properties measured at individual snowpits represented layer heterogeneity at the native airborne radar resolution (~2 metres), a 50 metres snow trench was excavated in March and April (Rutter et al., 2019). The measurements made at each snowpit (as described above) were repeated at 5 m intervals along the trench. Near-infrared (NIR) photos along the entire trench were collected for analysis of layer (dis)continuity.

**3.3.1    SAR imagery**

Trail Valley Creek flight lines were surveyed during both the March and April campaign periods. Figure 17 shows a mosaic of SnowSAR swathes measured during TVCEx-2 (Ku-band VV pol depicted).Distinct backscatter signatures associated with topography and vegetation were noted in the TVCEx mission data shown in Figure 17 as well as the other frequency and

polarization combinations. Where roughness tussock dominated surfaces were found in valley bottom environments, increased scattering was noted at both frequencies. Additionally, outcrops of shrub vegetation throughout the domain were associated

with        strong        scattering        where        interactions        with        the        snowpack        were        enhanced.

Figure 17: An example of SAR imagery (KuVV) over Trail Valley Creek during TVCEx-2. Location of weather stations
        indicated by purple dots.

### 3.3.2    Meteorological and snow conditions

Over the last 20 years, several meteorological stations were installed within TVC and its sub-basins. To supplement these
        stations, two additional sites were installed prior to the 2012/13 snow season in plateau and valley-bottom environments.
        Additional stations included sensors to monitor snow depth, snow temperature, soil temperature, and soil permittivity. The
        upland plateau station (179 m a.s.l.) was located on relatively flat, gently undulating terrain, covered almost entirely by
        gramanoid tundra. The valley bottom station (104 m a.s.l.) was surrounded by steep slopes along the northern valley boundary,





with tall shrub tundra found on south facing slopes and in proximity to nearby water features. Given the differences in the surrounding topography and vegetation, it was anticipated that the stations could be used to examine inter-basin differences in accumulation and snow transportation dynamics.

Figure 18 shows the seasonal progression of snow depth as observed at the upper plateau and valley bottom stations between

October 2012 and April 2013, in addition to air, snow and soil temperatures at the valley bottom station. While accumulation events common to both stations were identifiable (generally observed as step changes), the snowpack at the upland tundra station eroded during the early season due to sustained exposure to prevailing northwest winds. In comparison, terrain features sheltering the valley bottom allowed retention of early season snowfall, resulting in a snowpack that was over twice as deep by the December 2012 measurement campaign. A lack of standing vegetation in immediate proximity to the valley bottom

station moderated overall catch efficiency and therefore local retention of snowfall throughout the experiment. Storm events in early January and mid-February led to small snow depth increases at the upper tundra plateau station, eventually leading to similar snow depth between the two stations by the start of the March 2013 campaign. Three small snowfall events were observed at both stations between the March and April campaigns, which increased snow depth to over 30 cm by the end of the April field measurement campaigns. Wind redistribution of snow accumulation and shallow conditions observed at both

stations are typical of tundra environments where snow depth in open areas is generally limited to the height of the local standing vegetation (Derksen et. al., 2009).

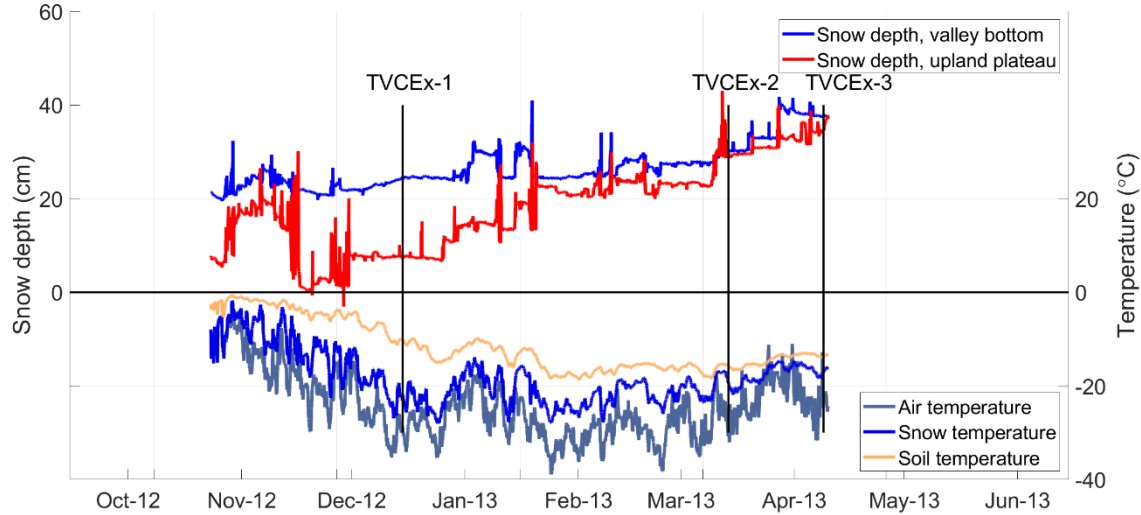



**Figure 18:** Snow depths observed at the Valley Bottom and Upland Plateau meteorological stations within the TVC basin. Air
temperature (200 cm height), snow temperature (10 cm height), and soil temperature (10 cm below surface) measured at the
Upper Plateau meteorological station. Approximate date of field campaigns and SnowSAR flights indicated with vertical lines.
No SnowSAR flight was performed during TVCEx-1.

Sustained cold air temperatures were observed at both meteorological stations throughout TVCEx. No melt events occurred,
with mean air temperatures well below -20°C during each of the campaigns. Due to sustained cold temperatures within the
snow volume, it is assumed that the radar observations were exclusively of dry snow. Soil temperatures at both stations were
also found to be well below 0°C (confirmed with soil pits completed at each meteorological station) and therefore assumed to
be frozen during all observation periods. For the vast majority of the winter, temperature gradients between the snow surface
and soil surface were greater than 20°C/m, strongly indicative of sustained periods of kinetic grain growth within the snow
volume. In tundra environments, such conditions generally result in the development of an early season basal depth hoar layer
and rapid faceting of wind slab layers (often referred to as relic or indurated wind slab, Sturm et al., 1997). Coupled with strong
winds and limited precipitation, the resulting snowpack throughout the domain is generally shallow, composed of contrasting
high-density wind slab and coarse-grained basal depth hoar components.  Observed differences in snow depth, and therefore
the vertical temperature gradient through the snowpack, also meant that strong spatial variations were evident in snow
microstructure and snow layer thickness (Rutter et al. 2019).
Observed snow depth and SWE from transects ranged from an average of 38 cm and 209 mm during TVCExp-1 in December
2012, to 51 cm and 248 mm in April 2013, respectively (Table 8). The variability in both snow depth and SWE was high,
being comparable to the Saariselkä site in Finland (Table 4).

Snow pits were characterized by two primary components: wind slab and basal depth hoar. The mean number of layers
observed in snow pits ranged between 5 and 7, with much of the variability associated with the number of wind slab layers.
Slab layers, composed of fine rounded grains, were a product of sustained wind events and subsequent mechanical deformation
of transported crystals. Slab thickness and layering was largely governed by proximity to standing vegetation or topographic
depositions whereby wind transported snow was able to accumulate as new layers.

In the early season, depth hoar layers formed in hollows between hummocks where fresh snow was sheltered. With subsequent
snowfall events, the height of open tundra depth hoar layers was found to increase with depth up to half the height of the total
snowpack. Where deeper snow was found (i.e. drifts), the depth hoar layer was limited, generally comparable in thickness to
layers found in shallower open tundra areas (< 30 cm).

Slab layers buried by sequential snow events were not immune to kinetic growth and from an early point in the season showed
signs of faceting. By April 2013, nearly the full volume of the open site snowpack was faceted (**Figure 19**). By April, faceted
slab layers, were composed of large, well bonded, cup like crystals with densities similar to their previous slab form (>250 kg



m$^{-3}$). These layers were easily distinguished from depth hoar where basal structures were poorly bonded with much lower densities (<250 kg m$^{-3}$).

**Table 8: Mean and standard deviation** of snow depth, SWE and density from Magnaprobe and SWE transects in TVC campaigns.

| Mission Id. | SD samples (#) | SD (cm) Mean, Std | SWE samples (#) | Density (kg m$^{-3}$) mean, std | SWE (mm) mean, std |
|---|---|---|---|---|---|
| TVCEx-1 | 7181 | 38, 19 | 251 | 209, 55 | 57, 41 |
| TVCEx-2 | 12009 | 53, 20 | 241 | 243, 49 | 113, 56 |
| TVCEx-3 | 22645 | 51, 19 | 402 | 248, 44 | 128, 58 |


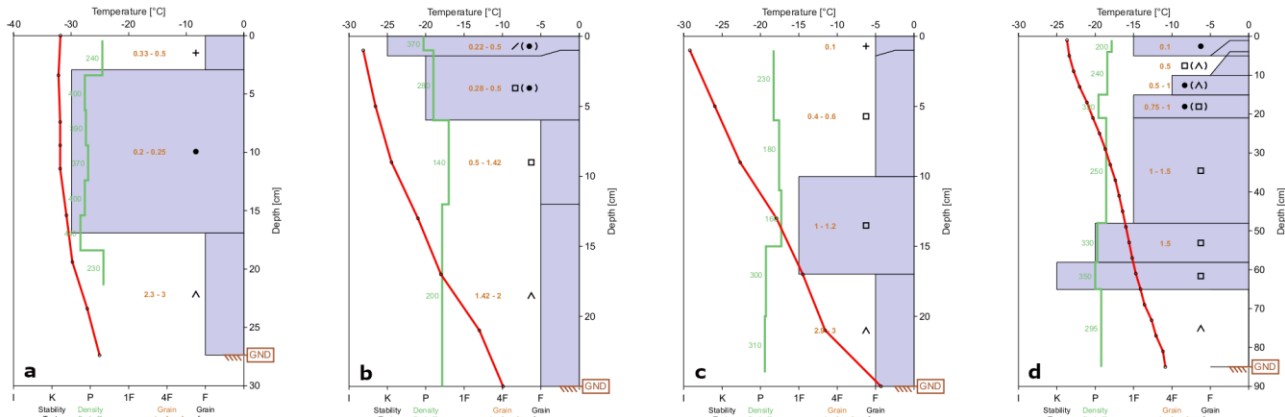

**Figure 19**: Examples of snow pits completed near the upland tundra meteorological during sequential TVCEx field campaigns (TVCEx-1: a,b. TVCEx-2: c TVCEx-3: d).

**4     Data preparation and classification**

**4.1     SnowSAR data processing**

Additional processing steps for SnowSAR data from NoSREx and TVCExp sites included calculation of  Universal Transverse Mercator (UTM) coordinates and local incidence angles (xyz files) for each backscatter pixel (code by Coccia, Trampuz, & Imbembo, 2011). These files were then processed in SAGA (Conrad et al., 2015), where the actual geocoded images in 2 m

spatial resolution and UTM projection were compiled. When re-sampling data from xyz-files to the 2x2m grids, 4-6 % of grid cells were left with no data, i.e. no observations were located in those grid cell areas. Correspondingly, 4-6 % of the grid cells





had two SnowSAR observations inside one grid cell. Cells without data were filled by calculating the mean value of the surrounding eight pixels, and for the grid cells with multiple measurements, the mean value of the two observations was calculated. These averaging operations during the data sampling decreased the standard deviation of the SnowSAR sigma
nought observations by approximately 1-1.5 %.

In the case of AlpSAR, MATLAB® code provided by Metasensing (Coccia, Trampuz, & Imbembo, 2011) was converted into Python to extract geometry information and backscatter maps from floating point data. Information from the aircraft, DEM and multilooked image files were used to receive xyz coordinates of each pixel to process the local incidence angle and sigma
nought maps and project them to a 2x2 m grid. Hence, the main difference between the AlpSAR data processing method and the Sodankylä, Saariselkä and TVC processing method was in the projection of the observations to the UTM-grid. For AlpSAR, the UTM-coordinates of the observations were calculated from the aircraft orbit information and a DEM, while for NoSREx and TVCExp, the UTM-coordinates were derived from the UTM corner coordinates of the images and heading information.

For AlpSAR, the resulting backscatter maps were compared with the scaled, geocoded maps provided by Metasensing (GTiff; scaled from 8bit to sigma nought (dB)) to verify the correct conversion with Python. For Sodankylä, Saariselkä and TVC, the image geometry was validated by comparing the resulted georeferenced grid data with other available geospatial data from the areas.

**4.2    Classification based on surface type**

Each SnowSAR grid cell was classified based on land cover classification (Table 9). At Sodankylä (boreal forest) and Saariselkä (tundra), land cover information from the national Finnish Corine2012 land cover database was applied (Härmä et al., 2013). Original Corine2012 classes were aggregated in order to reach the classification scheme. At TVC, land cover classification was based on an airborne LiDAR survey. At AlpSAR sites, land cover classification was based on public digital
databases of land cover (Digitaler Laser- und Luftbildatlas Tirol, Abteilung Geoinformation, Land Tirol). At the high Alpine site Rotmoos and glacier site Mittelbergferner, the mapping of the land cover extent was further supported by terrain corrected top-of atmosphere spectral reflectivity from geocoded Landsat images.

**Table 9:** Generalized land cover regimes for SnowSAR sites.

| Sodankylä | | |
|---|---|---|
| **LC id** | **Acronym** | **Description** |
| So1 | Fm | Coniferous/mixed forest on mineral soil |
| So2 | Fp | Coniferous/mixed forest on organic soil (peat) |
| So3 | We | Wetland (open bogs) |
| So4 | Me | Meadows, open fields (natural and cultivated) |





| So5 | Lr | Lakes and Rivers |
|---|---|---|
| So6 | O | Other (roads, buildings, barren; excluded from analysis) |
| **Saariselkä** | | |
| Sa1 | Tu | Tundra (non-vegetated) |
| Sa2 | Tl | Tundra- low shrub |
| Sa3 | Tt | Tundra- tall shrub |
| Sa4 | Fm | Coniferous/mixed forest on mineral soil |
| Sa5 | O | Other (lakes, rivers, wetlands, urban; excluded from analysis) |
| **Leutasch** | | |
| Le1 | Me | Meadows, open fields (natural and cultivated) |
| Le2 | Fo | Forest |
| Le3 | Sh | Shrub |
| Le4 | Wa | Water |
| Le5 | Ur | Urban (buildings; streets) |
| Le6 | Ro | Roads |
| Le7 | Ep | Electricity pylon |
| **Mittelbergferner** | | |
| Mf1 | If | Ice-free surfaces (rock, moraine) |
| Mf2 | Gi | Glacier ice-area |
| Mf3 | Gf | Glacier firn area |
| **Rotmoos** | | |
| Ro1 | Ag | Alpine grassland (sedges, grasses, scattered dwarf shrubs) |
| Ro2 | Sv | Sparse alpine vegetation (sedges, scattered tussocks) |
| Ro3 | Bo | Bog, marsh |
| Ro4 | RS | Rock, scree, moraine |
| Ro5 | Al | Alluvium |
| Ro6 | Wa | Water |
| Ro7 | Rb | Road, building |
| **Trail Valley Creek** | | |
| Tvc1 | Tu | Tundra (non-vegetated) |
| Tvc2 | Tl | Tundra- low shrub |
| Tvc3 | Tt | Tundra- tall shrub |
| Tvc4 | Fm | Coniferous/mixed forest on mineral soil |


## 4.3   Aggregation to 10 m resolution

SnowSAR images were resampled to 10 m resolution. The values of the output grid cells (10x10 m) were the mean value of
the calibrated sigma nought values (as natural numbers, not dB) in original 2x2 m grid cells. Sigma nought and local incidence

angle image mosaics composed of all swaths were compiled for each mission and each polarization. For each 10 m grid cell,
the standard deviation was calculated from all SnowSAR sigma nought observations inside the grid cell. Ancillary data were
resampled to the same grid as the SAR data (Table 10). A local DEM is available for each site as a separate layer, resampled
to the SAR grid resolution.





**Table 10:** Resampling methods for ancillary data.

| Dataset | Original resolution (m) | Output resolution (m) | Interpolation method |
|---|---|---|---|
| **Sodankylä/Saariselkä** | | | |
| DEM | 10 | 10 | Bilinear |
| Land cover | 20 | 10 | Nearest neighbor |
| Forest data | 20 | 10 | Nearest neighbor |
| **Leutasch** | | | |
| DEM | 10 | 10 | Bilinear |
| Land cover (shape files), Geo-Information Division, Gvt. Tyrol | 10 | 10 | Nearest neighbor |
| **Mittelbergferner** | | | |
| DEM | 10 | 10 | Bilinear |
| Landsat ETM images, resampled to 10 m (bilinear) | 10 (resampled) | 10 | Bilinear |
| **Rotmoos** | | | |
| DEM | 10 | 10 | Bilinear |
| Land cover (shape files, Gvt. Tyrol) | 10 | 10 | Nearest neighbor |
| Landsat ETM images, resampled to 10 m (bilinear) | 10 (resampled) | 10 | Bilinear |
| **Trail Valley Creek** | | | |
| DEM | 2 | 10 | Mean value |
| Vegetation cover | 2 | 10 | Modal value |

## 4.4 Content of data package and data formats

### 4.4.1 Gridded data

SnowSAR data are available for each site in a structured NetCDF database consisting of a layered 10 m spatial resolution geolocated matrix of: calibrated SAR backscattering intensity (sigma nought), land cover classification, DEM, and vegetation/forest data. Each channel of calibrated airborne data, KuVH, KuVV, XVH, XVV (radiometric calibration provided by Metasensing), forms the first layers of the dataset, with calibrated sigma nought, standard deviation and local incidence angle layers corresponding to each channel. The sigma nought and incidence angle data were resampled to the relevant resolution grid (10 m) by calculating the mean value of observations inside each grid cell. Tables 11-14 describe the content of the structured database separately for all sites.



**Table 11:** Content of structured database for Sodankylä.

| Layer | Content | Comments |
|---|---|---|
| 1-48 | SnowSAR $\sigma^0$ | Sigma nought for all missions in increasing order (00, 01, 02 … 10). Each mission with four layers: KuVH, KuVV, XVH, XVV. Missions include mission00 (March 2011) and mission04 with extended elevation angle |
| 49-96 | SnowSAR std | Standard deviation for all missions and polarizations |
| 97-144 | SnowSAR incidence angle | Local incidence angle for all missions and polarizations |
| 145 | LC2012 | Land cover type following Table 2 |
| 146 | CC | Canopy cover |
| 147 | TH | Tree height |
| 148 | VOL | Stem volume |
| 149 | DEM | Digital elevation model |

**Table 12:** Content of structured database for Saariselkä.

| Layer | Content | Comments |
|---|---|---|
| 1-8 | SnowSAR $\sigma^0$ | Sigma nought for missions T1 and T2, polarizations KuVH, KuVV, XVH and XVV |
| 9-16 | SnowSAR std | Standard deviation for missions T1 and T2, polarizations KuVH, KuVV, XVH and XVV |
| 17-24 | SnowSAR incidence angle | Local incidence angle for missions T1 and T2, polarizations KuVH, KuVV, XVH and XVV |
| 25 | LC2012 | Land cover type following Table 2 |
| 26 | CC | Canopy cover |
| 27 | TH | Tree height |
| 28 | VOL | Stem volume |
| 29 | DEM | Digital elevation model |


**Table 13:** Content of structured database for AlpSAR test sites (Leutasch, Rotmoos, Mittelbergferner).

| Layer | Content | Comments |
|---|---|---|





| 1-12 | SnowSAR $\sigma^0$ | Sigma nought (lin) for all missions (1, 2B, 3). Each mission with four layers: KuVH, KuVV, XVH, XVV. |
|---|---|---|
| 13-24 | SnowSAR std | Standard deviation for all missions and polarizations |
| 25-36 | SnowSAR incidence angle | Local incidence angle (deg) for all missions and polarizations |
| 37 | SCM | Land cover type following Table 2 |
| 38 | DEM | Digital elevation model |

**Table 14:** Content of structured database for TVC.

| Layer | Content | Comments |
|---|---|---|
| 1-8 | SnowSAR $\sigma^0$ | Sigma nought for missions M2 and M3, polarizations KuVH, KuVV, XVH and XVV |
| 9-16 | SnowSAR std | Standard deviation for missions M2 and M3, polarizations KuVH, KuVV, XVH and XVV |
| 17-24 | SnowSAR incidence angle | Local incidence angle for missions M2 and M3, polarizations KuVH, KuVV, XVH and XVV |
| 25 | VEG | Land cover type following Table 2 |
| 26 | DEM | Digital elevation model |

#### 4.4.2 Meteorological and snow data

Point based in situ measurements which could not be properly gridded, such as snow depth, snow density and SWE information are available separately as vector data in ESRI shapefiles (.shp). In this format, data can be geocoded to the same coordinate reference system as the gridded database. Snow pit information are provided as annotated .xls spreadsheets in a common format (one spreadsheet for each snow profile). Weather station data are provided as .csv files; measured variables vary based on available data and are indicated in file headers. The following tables give a summary of collected in situ data at each site.


**Table 15:** Summary of manual and automated datasets available from Sodankylä

| Measurement | Location | Parameters | Unit |
|---|---|---|---|
| Meteorological data | N 67.3618, E 26.6338 (forest clearing)<br>N 67.3669, E 26.6517 (wetland) | Air temperature<br>Dew point temperature<br>Wind speed | °C<br>°C<br>m/s |



| | | Wind gust | m/s |
|---|---|---|---|
| | | Pressure (sea and station level) | hPa |
| | | Present weather code | - |
| | | Height of lowest clouds | m |
| | | Total cloudiness | octa |
| | | Snow Depth | cm |
| | | Precipitation | mm |
| | | Global solar radiation | W/m² |
| | | Reflected solar radiation | W/m² |
| | | Soil dielectricity (5, 10, 20, 30, 40 cm depth) | - |
| | | Soil moisture (5, 10, 20, 30, 40 cm) | % vol |
| | | Soil Temperature (5, 10, 20, 30, 40 cm) | °C |
| | | Soil electric conductivity (5, 10, 20, 30, 40 cm) | dS/m |
| Snow pits | N 67.3620, E 26.6340 (forest clearing)  N 67.3670, E 26.6520 (wetland) | Snow stratigraphy  Hand hardness  Grain size (per layer)  Grain type (per layer)  Density (profile)  Temperature (profile) | cm  I, K, P, 1F, 4F, F  mm  Fierz et al. (2009)  kg/m³  °C |
| Snow transects | varies | SWE  Snow depth | mm  cm |

**Table 16:** Summary of manual and automated datasets available from Saariselkä

| Measurement | Location | Parameters | Unit |
|---|---|---|---|
| Meteorological data | N 68.3302, E 27.5506 | Air temperature  Soil temperature 5 and  10 cm depth (2 locations)  Snow depth | °C  °C    cm |
| Snow transects | varies | SWE  Snow depth | mm  cm |





**Table 17:** Summary of manual and automated datasets available from Leutasch

| Measurement | Location | Parameters | Unit |
|---|---|---|---|
| Meteorological data | N 47.3708, E 11.1425 | Relative Humidity | % |
| | | Air temperature | °C |
| | | Precipitation | mm |
| | | Snow depth | cm |
| | | Soil water content (3 and 7 cm) | % vol |
| | | Soil Temperature (3 and 7 cm) | °C |
| Snow pits | varies | Snow stratigraphy | cm |
| | | Hand hardness | I, K, P, 1F, 4F, F |
| | | Grain size | mm |
| | | Grain type | Fierz et al. (2009) |
| | | Density profile | kg/m$^3$ |
| | | Temperature profile | °C |
| Snow transects | varies | Snow depth | cm |

**Table 18:** Summary of manual and automated datasets available from Rotmoos

| Measurement | Location | Parameters | Unit |
|---|---|---|---|
| Meteorological data | N 46.8435, E 11.0236 | Relative Humidity | % |
| | | Air temperature | °C |
| | | Precipitation | mm |
| | | Snow depth | cm |
| | | Soil water content (3 and 7 cm) | % vol |
| | | Soil Temperature (3 and 7 cm) | °C |
| Snow pits | varies | Snow stratigraphy | cm |
| | | Hand hardness | I, K, P, 1F, 4F, F |
| | | Grain size | mm |
| | | | Fierz et al. |





| | | Grain type | (2009) |
| | | Density profile | kg/m³ |
| | | Temperature profile | °C |
| Snow transects | varies | Snow depth | cm |

**Table 19:** Summary of manual and automated datasets available from Mittelbergferner

| Measurement | Location | Parameters | Unit |
|---|---|---|---|
| Snow pits | varies | Snow stratigraphy | cm |
| | | Hand hardness | I, K, P, 1F, 4F, F |
| | | Grain size | mm |
| | | Grain type | Fierz et al. (2009) |
| | | Density profile | kg/m³ |
| | | Temperature profile | °C |
| Snow transects | varies | (seasonal) Snow depth | cm |


**Table 20:** Summary of manual and automated datasets available from Trail Valley Creek

| Measurement | Location | Parameters | Unit |
|---|---|---|---|
| Meteorological data | N 68.6933, W 133.6988 (Upland Plateau) | Relative Humidity | % |
| | | Air temperature | °C |
| | N 68.6873, W 133.6417 (Valley bottom) | Precipitation | mm |
| | | Snow depth | cm |
| | | Soil water content (3 and 7 cm) | cm |
| | | Soil Temperature (3 and 7 cm) | °C |
| Snow pits | varies | Snow stratigraphy | cm |
| | | Hand hardness | I, K, P, 1F, 4F, F |
| | | Grain size | mm |
| | | Grain type | Fierz et al. (2009) |
| | | Specific Surface Area | m²/kg |
| | | Density profile | kg/m³ |




| | | Temperature profile | °C |
|---|---|---|---|
| Snow transects | varies | SWE | mm |
| | | Snow depth | cm |

## 5    Data availability

Data are available via: https://doi.pangaea.de/10.1594/PANGAEA.933255 (Lemmetyinen et al. 2021). Additionally, original

SnowSAR    data    are    available    separately    via    the    ESA    Campaign    data    portal
(https://earth.esa.int/eogateway/campaigns/snowsar-nosrex-tvcexp-and-alpsar ).

NOTE: A temporary link to access the data without login information is provided for reviewers of this manuscript:
https://www.pangaea.de/tok/e8c562c3c8a15ac34daa83d00c76fcb347330884

## 6    Considerations for use of data

### 6.1    Co-locating in situ observations with SnowSAR backscatter

The goal of snow measurements during SnowSAR missions was to provide a co-located reference of snow conditions to the
SnowSAR swaths. For some missions, some of the snow measurements were made outside of final SnowSAR swaths due to
uncertainty in aircraft navigation, the cancelling of some tracks due to limited flight time, and limited swath width. Figure 20

shows an example for NoSREx M02 in Sodankylä, where co-location of SnowSAR data and in situ snow measurements was
poor; measurements of snow depth (red dots) and SWE (green dots) are shown against the measured XVV backscatter. Several
snow measurement transects are seen to lie outside of the coverage of SnowSAR swathes, making opportunities for direct
comparison of backscatter with SWE and snow depth limited. However, in these cases, snow measurements can still be used
as a reference for changes in snow conditions between missions, e.g. characterising different types of land cover (Hannula et

al., 2016).

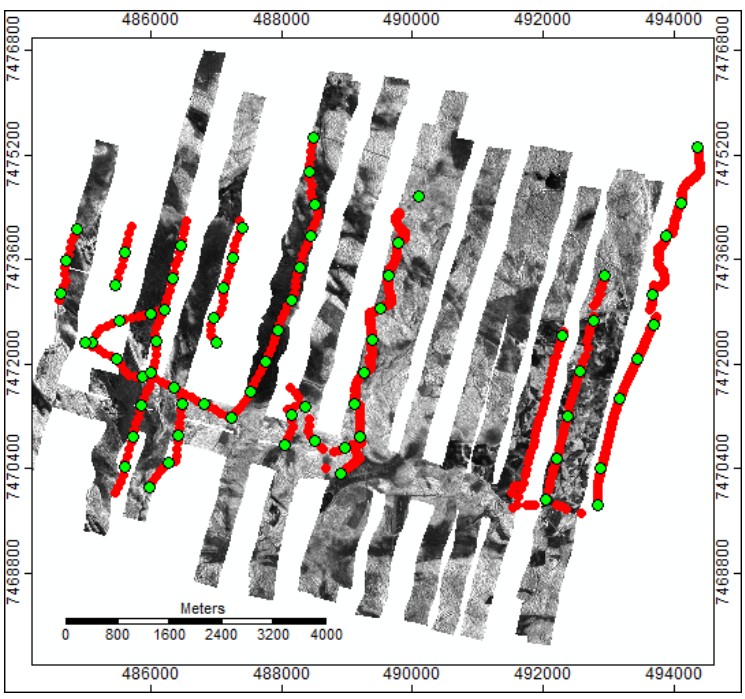

**Figure 20.** SnowSAR X-band VV pol backscatter for M02 in Sodankylä. Location of in situ snow measurements depicted with red (snow depth) and green (SWE) dots.

## 6.2 Effect of incidence angle

SnowSAR data are provided with an effective incidence angle relative to ground level. Due to the low flight altitudes compared to swath width, the incidence angle across-track varies more than for typical space-borne SAR instruments. To ensure proper focusing and calibration of the swaths, the nominal incidence angle range is limited from 35 to 45 degrees. There is still a notable effect on backscatter intensity across the swath. Figure 21 provides an example of the angular dependence of backscatter for NoSREx mission T02 at the Saariselkä test site for four land cover categories. A monotonically decreasing trend of backscattering intensity with increasing incidence angle is evident for all SnowSAR channels and most land cover types (an exception is the Tt (Tundra – tall shrub) land cover class at XVH and KuVH); this type of effect can be expected for surfaces where surface scatter occurs mostly in a specular direction. However, the angular dependence is not similar for all missions or land cover types. **Figure 22** depicts the angular dependence of backscatter for M02 in Sodankylä. Here, a clearly decreasing trend is only evident for the XVH channel. KuVV and XVV show no obvious change with incidence angle, while the KuVH channel shows a distinct increasing trend, indicating that volume scattering could be the dominant contributor to the overall intensity of backscatter.

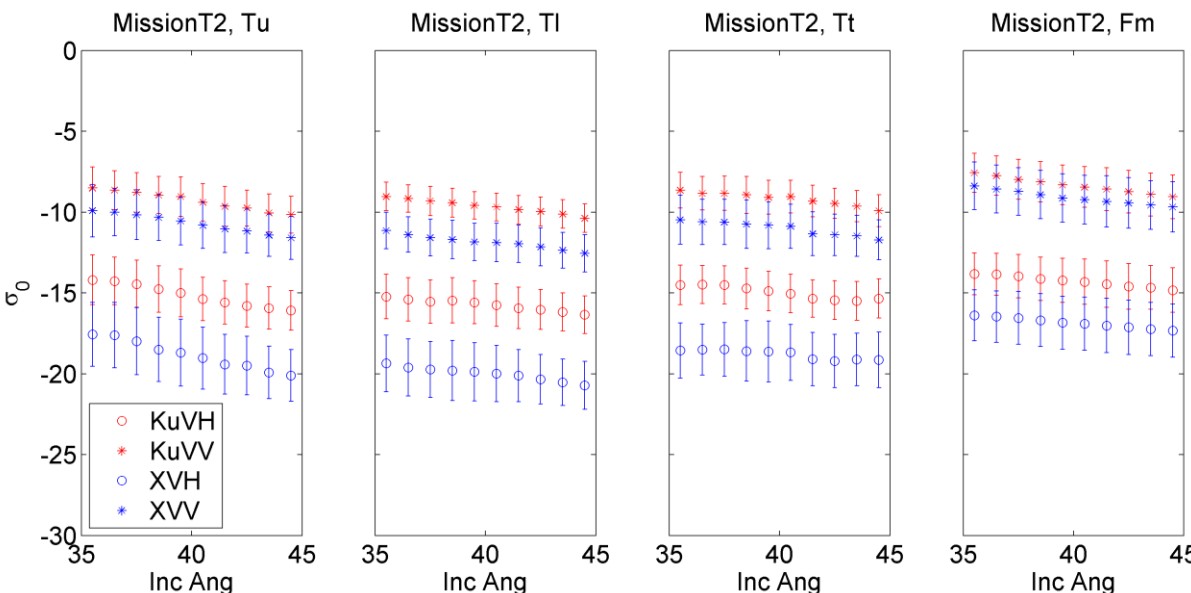

**Figure 21:** Sensitivity of backscatter (in dB) to incidence angle during SnowSAR mission T02 for main land cover types.

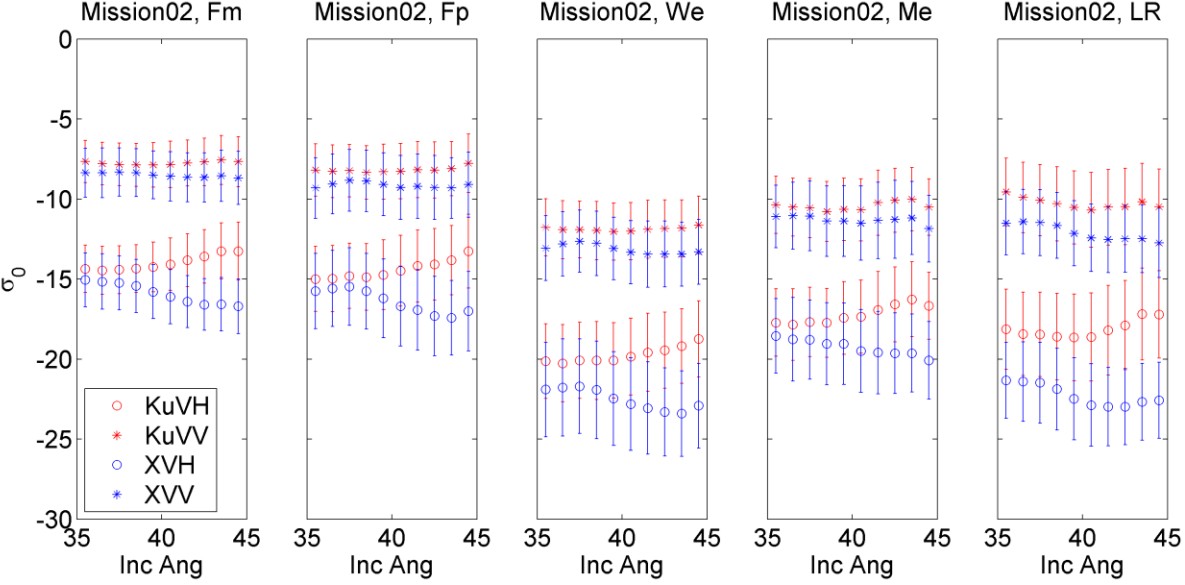

**Figure 22:** Sensitivity of backscatter (in dB) to incidence angle during SnowSAR mission M02 for main land cover types.

# 7    Summary

Because of high resolution, independency from cloud cover and sun illumination, and sensitivity to snow properties (depending on frequencies), radar measurements are an attractive option for addressing the spatial and temporal measurement requirements



related to the monitoring of snow water equivalent. In recent years new mission development (e.g. Rott et al., 2010) has focused on frequencies like X- and Ku-band which are sensitive to SWE through volume scattering. The backscattering properties of

snow at these frequencies have been studied using ground-based systems, allowing the development and testing of forward models and basic retrieval approaches (King et al., 2014; Lemmetyinen et al., 2018). Nevertheless, the limited area observed by ground-based sensors cannot be representative of the natural variability of snow over a certain region. Airborne datasets are required to advance this potential by supporting the evaluation of radar modeling capabilities for variable land cover such as vegetation (e.g. Montomoli et al., 2015) and development of new inversion approaches at larger spatial scales, but these

data are extremely limited. The SnowSAR dataset represents the first collective effort by the international snow community to compile airborne radar, coincident snow observations, and the required ancillary datasets in a harmonized way from a network of sites which capture a wide range of snow and land cover conditions. The airborne data, as well as parts of the collected snow parameters, have already been applied to study the backscatter characteristics of snow cover (e.g. Zhu et al., 2018; King et al., 2018; Rutter et al., 2019) and vegetation (Cohen et al., 2015; Montomoli et al., 2015), and the statistical variability of

snow cover characteristics (Hannula et al., 2016). The relatively large amount of collected data has enabled to study also the use of machine-learning approaches for SWE retrieval (Santi et al., 2021). Lessons learned from the SnowSAR campaigns have informed planning and execution of subsequent field experiments, and will contribute to more consistent and coordinated data acquisition and analysis within the snow remote sensing community.

**Author contribution**

J.L, A.K, J.V., H-R.H., S.S., H.R., T.N., E. R., K.E., H.-P. M., R. F., M. S. A., C. D., J. K. and N. R. took part in field data sampling as well as compilation and analysis of airborne and field datasets. A. M. and A. C. took part in airborne data collection and calibration. J. C., I. M., M. S., G. M., E. S., M. L.-L., R. E., C. M. and M. K. took part in compilation and analysis of the airborne and field datasets. All authors

provided comments and contributions to the text and figures in the manuscript.

**Acknowledgements**

Staff at FMI as well as Martin Proksch, Martin Schneebeli, Matthias Jaggi (WSL Institute for Snow and Avalanche Research

SLF), Mark Dixon (NASA), Andreas Wiesmann, Christian Mätzler (Gamma Remote Sensing) and Maria Gritsevich (Finnish Geodetic Institute) are acknowledged for participating in field work in Sodankylä. The AlpSAR field activities were a joint effort of numerous colleagues from ENVEO IT, Innsbruck, Austria; BFW, Department of Natural Hazards, Innsbruck, Austria, U.S. Department of Agriculture Forest Service, Fort Collins, CO, USA, and Department of Geoscience, Boise State University, Boise, ID, USA. Rainer Prinz (presently at University of Innsbruck) is acknowledged for major efforts in the preparation and

field activities of the AlpSAR campaigns. Arvids Silis, Peter Toose and Bennoit Montpetit (Environment and Climate Change Canada, ECCC), Chris Larsen and Mathew Sturm (University of Alaska Fairbanks), Chris Hiemstra and Art Galvin (Cold



Regions Research and Engineering Laboratory), Glen Liston (Colorado State University) and Tom Watts (Northumbria University) are acknowledged for participating in field work at TVC. Thanks to Philip Marsh (Wilfrid Laurier University), Cuyler Onclin, and Mark Russell (ECCC) for logistical support.


SnowSAR development and deployments at all sites were supported by the European Space Agency (ESA) (Contract No. 4000101697/10/NL/FF/ef). Field work at Sodankylä as well as data analysis at all sites was supported by ESA (Contract Nos. 22671/09/NL/JA and. 4000118400/16/NL/FF/gp). The field activities for AlpSAR were supported by ESA (Contract No. 4000107780 /13/NL/BJ/lf), by the Austrian Space Application Program (ASAP-7) Contract No. 828345, and by NASA / JPL-

CalTech, Pasadena, CA, USA. Field activities at TVC were supported by ESA, University of Alaska Fairbanks, NASA (NNX13AQ90G and NNX15AC09G), Canadian Space Agency (13MOA07103), and Environment and Climate Change Canada.

Snowpit illustrations were drawn using niViz (https://niviz.org).

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
