# Peer review of "Airborne SnowSAR data at X- and Ku- bands over boreal forest, alpine and tundra snow cover"

_Earth System Science Data, 2021_

## Referee Comment (RC1)

**Airborne SnowSAR data at X- and Ku- bands over boreal forest, alpine and tundra snow cover**

**Authors**: Juha Lemmetyinen[1], Juval Cohen[1], Anna Kontu[1], Juho Vehviläinen[1], Henna-Reetta Hannula[1], Ioanna Merkouriadi[1], Stefan Scheiblauer[2], Helmut Rott[2], Thomas Nagler[2], Elisabeth Ripper[2], Kelly Elder[3], Hans- 5 Peter Marshall[4], Reinhard Fromm[5], Marc S. Adams[5], Chris Derksen[6], Joshua King[6], Adriano Meta[7], Alex Coccia[7], Nick Rutter[8], Melody Sandells[8], Giovanni Macelloni[9], Emanuele Santi[9], Marion Leduc- Leballeur[9], Richard Essery[10], Cecile Menard[10] and Michael Kern[11]

**Overall Recommendation**: Major revision.

**General Statement**

This paper well describes historical deployments of SnowSAR at X- and Ku- band for airborne active microwave observations for SWE retrieval from remote sensing.

1. While worthwhile to archive the past applications of the airborne SnowSAR deployments, it would be desirable to point out lessons learned from expensive airborne campaign along with in-situ snow and weather observations on the ground. Please refer to point-by-point conclusions in the past literature similar to:

Mätzler, Christian, and Erwin Schanda. "Snow mapping with active microwave sensors." *Remote Sensing* 5.2 (1984): 409-422.
Matzler, Christian, Erwin Schanda, and Walter Good. "Towards the definition of optimum sensor specifications for microwave remote sensing of snow." *IEEE Transactions on Geoscience and Remote Sensing* 1 (1982): 57-66.
Foster, J. L., et al. "Derivation of snow water equivalent in boreal forests using microwave radiometry." *Arctic* (1991): 147-152.

2. Another limitation is in a lack of contributions from snow hydrology models such as SNOWPACK and CROCUS. Please include how applications of the snow hydrology model can support SWE retrieval algorithm of using SnowSAR, i.e. microwave volume scattering approach.
3. Another note could be made with in-situ observations not limited to snowpit measurements but including ground-based remote sensing measurements. Recently, state-of-art ground technologies have been proposed including Specific Surface Area, Tomography Scanning of snow microstructure, and ground-based remote sensing measurements. I think an inclusion of the recent development of field and laboratory technologies would make synergy with airborne SnowSAR observations toward SWE retrieval algorithms.

4. While this paper is aimed at summarizing SnowSAR airborne observations, it would be useful to indicate a brief future planning how to use SnowSAR to retrieve SWE at the end. For example, 'background scattering' is quite well known, and the paper also summarizes the lower boundary scattering. A paragraph or a diagram would benefit the audience to understand how the SnowSAR observations and ancillary dataset will be utilized for SWE retrieval.

**Minor issue**

1. ***In abstract and line 55:*** 'dual polarized (VV/VH) ➔ dual polarized (VV, VH, HV, and HH). Is there any physical reason only using VV, VH, and HV, not HH? If so, please provide this in the beginning.

2. 'operable from a small aircraft' ➔ 'operated by various sizes of aircrafts'. It was deployed by P3 back in 2017 at NASA SnowEx

3. Any reference for 'In Canada, the TVCEx campaign took place in March and April 2013, with two flight campaigns over sites in the Trail Valley Creek (TVC) watershed, Northwest Territories, representative of the tundra snow regime.'? I found Di Leo, D., et al. "Radiometric calibration of the SnowSAR images of sub-artic open tundra watershed in Canada." (2015): 7-7.

4. Figure 1 Caption: 'Location of weather station' ➔ 'Location of weather station, ground-based remote sensing, and in-situ snowpit observations' to be complete

5. Figure 1 north and south: the left panel may be 90 degree counter clockwise rotation to satisfy the right panel. Try to be physically correct the aerial photo along with vegetation map. It will help the retrieval algorithm to account for vegetation effect on microwave volume scattering.

6. Figure 4: It is excellent to see flight occurrences such as M00 to M10. It may be helpful to move y-axis of air temperature up not to avoid to see SWE evolution.

7. Line 235: '17 March 2011 (M00)' ➔ It is helpful to have local time to interpret a diurnal status of snowpack during a daytime.

8. 'SnowSAR mission T1, T2' also needs local time, not the UTC.

9. Specify which frequency and polarization in Figure 3 and 10.

10. Figure 1, 2, etc: Please consider 'google mapTM' embedded format.

11. Spatial distribution of snowpit observations: For a microwave forward modeling perspective, locations of snowpits are essential to be compared with SnowSAR. I think a map of spatial distribution of snowpits is prerequisite at least for one or two campaigns.

---

## Author Response (AR1)

Response to RC1 comments on **Airborne SnowSAR data at X- and Ku- bands over boreal forest, alpine and tundra snow cover**

**RC1:**
*This paper well describes historical deployments of SnowSAR at X- and Ku- band for airborne active microwave observations for SWE retrieval from remote sensing.*

Response:
We thank the reviewer for his constructive comments and observations. We have done our best to comply to the suggested edits and additions. Where this was not done, we have provided justified answers below.

*RC1:*
*1. While worthwhile to archive the past applications of the airborne SnowSAR deployments, it would be desirable to point out lessons learned from expensive airborne campaign along with in-situ snow and weather observations on the ground. Please refer to point-by-point conclusions in the past literature similar to:*
*Mätzler, Christian, and Erwin Schanda. "Snow mapping with active microwave sensors." Remote Sensing 5.2 (1984): 409-422.*
*Matzler, Christian, Erwin Schanda, and Walter Good. "Towards the definition of optimum sensor specifications for microwave remote sensing of snow." IEEE Transactions on Geoscience and Remote Sensing 1 (1982): 57-66.*
*Foster, J. L., et al. "Derivation of snow water equivalent in boreal forests using microwave radiometry." Arctic (1991): 147-152.*

Response:
Thank you, this is a good point. We have included a new "lessons learnt" section which summarizes our experiences in collecting airborne radar data for snow cover. However, as this paper is meant as a presentation of the data, we refrain of making any conclusions as to e.g. the potential merits of the collected radar data and the volume scattering approach for SWE retrieval. The "lessons learnt" are thus meant as a technical reference of experiences (and mistakes!) when organizing possible future campaigns.

*RC1:*
*2. Another limitation is in a lack of contributions from snow hydrology models such as SNOWPACK and CROCUS. Please include how applications of the snow hydrology model can support SWE retrieval algorithm of using SnowSAR, i.e. microwave volume scattering approach.*

Response:
Thank you, good point, we have included a mention of this in section 6.1, also referencing recent work.

*"Furthermore, physical snow models can be applied to fill spatiotemporal gaps in the SnowSAR in situ data, supporting forward modelling or retrieval studies (see e.g. Liston &Elder, 2006; Merkouriadi et al., 2021)."*

*RC1:*
*3. Another note could be made with in-situ observations not limited to snowpit measurements but including ground-based remote sensing measurements. Recently, stateof-art ground technologies have been proposed including Specific Surface Area, Tomography Scanning of snow microstructure, and ground-based remote sensing measurements. I think an inclusion of the recent development of field and laboratory technologies would make synergy with airborne SnowSAR observations toward SWE retrieval algorithms.*

Response:
Thank you. We have added a more clear mention of the more advanced techniques for snow microstructure quantification and their potential benefit in the "lessons learnt" -section with references. Indeed, many of these techniques were applied during the campaigns, e.g. SSA measurements are available from most sites, and are available on separate request.

Coincident ground-based remote sensing was available for the Finnish dataset. A mention of these is now included with referencing.

*RC1:*
*4. While this paper is aimed at summarizing SnowSAR airborne observations, it would be useful to indicate a brief future planning how to use SnowSAR to retrieve SWE at the end. For example, 'background scattering' is quite well known, and the paper also summarizes the lower boundary scattering. A paragraph or a diagram would benefit the audience to understand how.*

Response:
This is a good point; a new paragraph is added to the Section 6.3 (lessons learnt), regarding the use of the data for SWE retrieval.

*"With the exception of Leutasch for AlpSAR-1, deployment of SnowSAR was too late for capturing a snow-free scene (despite best efforts). This poses a problem for testing e.g. the CoReH2O retrieval approach which compares backscatter from snow cover to an earlier snow-free surface. When testing potential retrieval approaches for other sites than Leutasch, users may have to resort to simulations for generating a snow-free scene of X and Ku band backscatter. For any future campaign testing a similar retrieval approach to CoReH2O, acquiring a reference radar image in snow free conditions should be a priority."*

*RC1:*
**Minor issue**
1. **In abstract and line 55:** *'dual polarized (VV/VH)* ➔ *dual polarized (VV, VH, HV, and HH). Is there any physical reason only using VV, VH, and HV, not HH? If so, please provide this in the beginning.*

Response:
The SnowSAR instrument, while technically capable of all four orthogonal transmit/receive modes, was only operate in VV and VH pol modes during the campaigns in Finland, Austria and Canada. Therefore, we would prefer to keep the original notation of VV/VH.

*RC1:*
*2. 'operable from a small aircraft' ➔ 'operated by various sizes of aircrafts'. It was deployed by P3 back in 2017 at NASA SnowEx*

Response:
Agreed, corrected.

*RC1:*
*3. Any reference for 'In Canada, the TVCEx campaign took place in March and April 2013, with two flight campaigns over sites in the Trail Valley Creek (TVC) watershed, Northwest Territories, representative of the tundra snow regime.'? I found*
*Di Leo, D., et al. "Radiometric calibration of the SnowSAR images of sub-artic open tundra watershed in Canada." (2015): 7-7.*

Response:
The best reference here is King et al., 2018, which is peer-reviewed. However, we would not like to place references in the abstract. King et al. (2018) is cited in several places in the text.

*RC1:*
*4. Figure 1 Caption: 'Location of weather station' ➜ 'Location of weather station, groundbased remote sensing, and in-situ snowpit observations' to be complete*

Response:
We would prefer to keep the original legend for the following reasons: 1) ground-based remote sensing are not a part of the dataset 2) regular snow pit observations were made at two locations, neither of which correspond to the weather station. The coordinates of the snow pit observations are included in the dataset.

*RC1:*
*5. Figure 1 north and south: the left panel may be 90 degree counter clockwise rotation to satisfy the right panel. Try to be physically correct the aerial photo along with vegetation map. It will help the retrieval algorithm to account for vegetation effect on microwave volume scattering.*

Response:
The left panel in the figure is an aerial photograph meant to give a general idea of the typical scenery, and is not intended to cover the entire test site. Similar photos are included for all sites. This has been clarified in all the figure captions. The essential data for volume scattering analysis is included in the dataset itself on e.g. forest characteristics.

*RC1:*
*6. Figure 4: It is excellent to see flight occurrences such as M00 to M10. It may be helpful to move y-axis of air temperature up not to avoid to see SWE evolution.*

Response:
Agreed. We adjusted the axis for all figures to minimize overlap of SD and temperature data.

*RC1:*
*7. Line 235: '17 March 2011 (M00)' ➜ It is helpful to have local time to interpret a diurnal status of snowpack during a daytime.*

Response:
This is a good point. We added the start and end times of data acquisitions in summary Tables 3, 5, and 7. Note that 'M00' refers to the mission id, not time (not sure if this was implied).

*RC1:*
*8. 'SnowSAR mission T1, T2' also needs local time, not the UTC.*

Response:
See response above.

*RC1:*
*9. Specify which frequency and polarization in Figure 3 and 10.*

Response:
Thank you for noting this. All examples are KuVV. Added mention in caption of Figure 3, which was missing.

*RC1:*
*10. Figure 1, 2, etc: Please consider 'google mapTM' embedded format.*

Response:
Here, we were unsure what the reviewer meant. However, all the data fields, including the land cover maps depicted in Figures 1 and 2, are available in the database with embedded geocoding, enabling the user to easily apply them in e.g. google maps and other GIS software.

*RC1:*
*11. Spatial distribution of snowpit observations: For a microwave forward modeling perspective, locations of snowpits are essential to be compared with SnowSAR. I think a map of spatial distribution of snowpits is prerequisite at least for one or two campaigns.*

Response:
We agree the location of Snowpit data is essential for colocation with SnowSAR data; therefore, coordinates of the snowpits are always included in the dataset. However, in particular for TVC the number and location of snowpits was large and the location of the pits varied during the campaigns; therefore the addition of the location e.g. over land cover maps would be cumbersome due to the large number of individual locations, and we feel this would not provide any added value. Nevertheless, as an example, we have added the location of the regular Snowpits in Sodankylä, Finland, on Figure 20, which exemplifies the difficulty of finding collocated snow and radar information for some of the campaigns.

Response to RC2 comments on **Airborne SnowSAR data at X- and Ku- bands over boreal forest, alpine and tundra snow cover**

*RC2:*
*The paper is a report on the data collected by SnowSAR. It is a report that gives the details of the sites, the SAR imagery and the ground truth. The report covers the essentials of the data*
*I suggest the following major revisions that will be useful for the readers and users of the data.*

We thank the reviewer for the comments. We have either modified the paper according to the provided suggestions or have provided a justification where we think our original approach was more appropriate.

*RC2: It is well known that the microstructure such as grain size, SSA have a significant influence on the sigma0. It has more influence than snow density. The paper can give more weight to plotting and showing the results of grain size. The figure label on grainsize are hard to read. Please also draw curves to show the change of grain size with depth. Adding grain sizes to the tables of snow depth and swe, densities will also help.*

Response: We agree the labels were difficult to read; we reduced the amount of figures in a row, which hopefully makes these easier to read. The figures were created using interactive snow profile visualizer tool (https://run.niviz.org/ ), depicting the output of .CAAML-format files which form part of the dataset. Unfortunately, Niviz does not allow to plot grain size as a top-to-bottom curve in the "mobile profiles" we have generated as examples. Nor does it allow to adjust the font size. We will try to ascertain the figures are readable in the final manuscript.

*RC2 Snow wetness will dramatically reduce the usefulness of the approach. A table describing the wetness of each of the mission and the areas should be provided.*

Response: Every mission was flown in dry snow conditions, with the possible exception of Mission 01 for SnowSAR flights in Finland, where snowpit data indicated some residual moisture in the snow. This has been explicitly mentioned in the text:

*"During M01 (**Error! Reference source not found.**b), the lower layers of the snowpack were at 0°C, indicating the presence of liquid water, which was likely to affect observed backscatter for that date."*

We added also a mention in Table 3 on wet snow conditions during M01.

*RC2: There are more than 10 missions in Finland which are supposed to give a time series of sigma0. Since there are many parameters in swe retrieval, times series measurements will enhance the retrieval accuracy. However, the time series concept is not emphasized in the paper. The authors can add the time series evolution of microstructure, grain sizes, snow densities to figures 4 and 5.*

Response: Thank you for the suggestion. The Finnish dataset indeed provides a time series aspect. As suggested, we have added figures providing the mean grain size and density, together with discussion (Figure 7 in revised manuscript).

*RC2: There will be many questions from users on sigma0, ground truth data etc. The section on data availability is too short. The authors should provide the contact person among the authors such as whom to contact for questions on Finland data, Alps data, TVC data etc.*

The contact people were included already in the "correspondence to" section beneath the abstract. However, we agree it is a good idea to add these in the Data availability section also.

*RC2: The sigma0 in the original AlpSAR data that is available from the ESA portal is not normalized. It would be helpful if the authors could add a short section on what the signal amplitude values mean in the original SnowSAR geotiff data. It will be useful for users who might be working with the geotiff amplitude files.*

In our work we did not use the GeoTIFF files distributed via the ESA server. Instead, we applied the radiometrically calibrated observation data (beta0) together with imaging geometry and topography information, from which the sigma0 is calculated (matlab code provided by MetaSensing). These data are in the ".dat" files on the ESA server.

The purpose the published dataset and this adjoining paper is specifically to make the SnowSAR data more easily accessible. In this context, we refer users to contact MetaSenging if they prefer to use the original raw data. A note of this has been added to the "data availability" section and the "correspondence to" section (see comment above).

*RC2 :The temporary link to access the gridded data (https://www.pangaea.de/tok/e8c562c3c8a15ac34daa83d00c76fcb347330884) in section 5 (page 43)does not work. If the authors could fix that, it will be helpful to users who need access to the final data set.*

We apologize. Apparently the anonymous, login-free link provided by PANGAEA to reviewers stopped working during the review process. The final link (requiring login) does work.

---

## Referee Report (RR1)

Comments on the paper **Airborne SnowSAR data at X- and Ku- bands over boreal forest, alpine and tundra snow cover**

I accept the revisions